# The Impact of a Simple Representation of Non-Structural Carbohydrates on the Simulated Response of Tropical Forests to Drought

**Simon Jones**[1], **Lucy Rowland**[2], **Peter Cox**[1], **Debbie Hemming**[3], **Andy Wiltshire**[3], **Karina Williams**[3,4], **Nicolas C. Parazoo**[5], **Junjie Liu**[5], **Antonio C. L. da Costa**[6], **Patrick Meir**[7,8], **Maurizio Mencuccini**[9,10], **and Anna Harper**[1]

[1]College of Engineering, Mathematics and Physical Sciences, University of Exeter, Exeter, Devon EX4 4QF, UK
[2]College of Life and Environmental Sciences, University of Exeter, Exeter, Devon EX4 4QF, UK
[3]Met Office Hadley Centre, FitzRoy Road, Exeter EX1 3PB, UK
[4]Global Systems Institute, University of Exeter, Laver Building, North ParkRoad, Exeter EX4 4QE
[5]California Institute of Technology, Jet Propulsion Laboratory, 4800 Oak Grove Drive, Pasadena, CA 91109
[6]Instituto de Geosciencias, Universidade Federal do Para, Belem, Brazil
[7]Research School of Biology, Australian National University, Canberra ACT 2601 Australia and
[8]School of Geosciences, University of Edinburgh, Edinburgh EH9 3FF UK
[9]ICREA, Pg. Lluís Companys 23, 08010 Barcelona (Spain)
[10]CREAF, Universidad Autonoma de Barcelona, Cerdanyola del Valles 08193, Barcelona (Spain)

**Correspondence:** Simon Jones (sj326@exeter.ac.uk)

**Abstract.** Accurately representing the response of ecosystems to environmental change in land surface models (LSM) is crucial to making accurate predictions of future climate. Many LSMs do not correctly capture plant respiration and growth fluxes, particularly in response to extreme climatic events. This is in part due to the unrealistic assumption that total plant carbon expenditure (PCE) is always equal to gross carbon accumulation by photosynthesis. We present and evaluate a simple model of labile carbon storage and utilisation (SUGAR), designed to be integrated into an LSM, that allows simulated plant respiration and growth to vary independently of photosynthesis. SUGAR buffers simulated PCE against seasonal variation in photosynthesis, producing more constant (less variable) predictions of plant growth and respiration relative to an LSM that does not represent labile carbon storage. This allows the model to more accurately capture observed carbon fluxes at a large-scale drought experiment in a tropical moist forest in the Amazon, relative to the Joint UK Land Environment Simulator LSM (JULES). SUGAR is designed to improve the representation of carbon storage in LSMs and provides a simple framework that allows new processes to be integrated as the empirical understanding of carbon storage in plants improves. The study highlights the need for future research into carbon storage and allocation in plants, particularly in response to extreme climate events such as drought.

---

## 1 Introduction

Correctly representing the balance between plant photosynthesis, growth and autotrophic respiration in the land surface model (LSM) component of Earth System Models (ESMs), is crucial to making accurate projections of global climate in the future.

Forests cover nearly 4000 Mha (UN Food and Agriculture Organization Rome, 2015) of the worlds land surface and represent a significant sink of carbon from the atmosphere, sequestering $2.4 \pm 0.4 \, \mathrm{PgCyr}^{-1}$, roughly 25% of total annual anthropogenic carbon emissions (IPCC, 2014). Most LSMs simulate growth and respiration as equal to instantaneous photosynthesis (Fatichi et al., 2014). Consequently at any given time, the total rate of carbon utilisation by respiration and growth, referred to as plant carbon expenditure (PCE), is equal to the rate of carbon accumulation by photosynthesis, commonly referred to as gross primary productivity (GPP). However, in reality growth and respiration are not so strictly coupled to photosynthesis and plants regularly experience periods when the supply of carbon from photosynthesis does not equal the demands of growth and respiration (Körner, 2003; Muller et al., 2011). This asynchrony between supply and demand is facilitated by reserve pools of labile carbon known collectively as non-structural carbohydrates (NSCs). The NSC pool within a plant accumulates when photosynthesis exceeds carbon demand and is drawn upon to sustain growth and respiration when they are not supported by instantaneous photosynthetic assimilation (Hartmann and Trumbore, 2016; Dietze et al., 2014). NSCs therefore act as a buffer, allowing key functional processes to be maintained, even when photosynthetic accumulation is low. This buffering is particularly important during periods of environmental stress, which can lead to reduced productivity over seasonal to multi-annual time-scales. During prolonged periods of stress, carbon utilisation rates can diverge significantly from photosynthesis (Metcalfe et al., 2010; Doughty et al., 2015b, a)implying that plants rely heavily on their NSC reserves during these periods. Without simulating NSC storage LSMs remain unable to capture this asynchrony between GPP and PCE and so fail to correctly simulate forest level respiration and growth fluxes.

The ability to sustain respiration and growth during periods of reduced productivity is an important process that can allow plants to survive, and recover from extreme short-term climate events, such as drought (Doughty et al., 2015b). Under low water availability the transport of water from roots to other organs can be compromised by the cavitation of xylem tissue in the plant (Martínez-Vilalta et al., 2014; Sperry and Love, 2015; Tyree and Sperry, 1989). Xylem damage can lead to a drop in hydraulic conductance, resulting in damage to plant tissue and increased risk of mortality (Rowland et al., 2015; Anderegg and Anderegg, 2013; McDowell et al., 2008). Plants combat this threat through control over the aperture of their stomata. Closing the stomata reduces water loss through transpiration and lowers the risk of xylem damage and hydraulic failure. The trade-off to this strategy is a reduction in productivity. The ability of a plant to employ this strategy is therefore reliant on its ability to store and utilise NSC. If carbon demand exceeds supply over long periods of drought, NSC reserves become exhausted, causing essential elements of plant function to fail, a process termed 'carbon starvation'. Carbon starvation and hydraulic failure are tightly linked processes (Mitchell et al., 2013; Adams et al., 2017), not only because of their shared dependence on stomatal conductance, but also due to the role that carbohydrates have in processes such as osmoregulation and potentially in refilling of embolised xylem (Sevanto et al., 2014). Carbon starvation may accelerate the effects of hydraulic failure and in some cases, itself lead directly to mortality (Galiano et al., 2011; Adams et al., 2013). Recent developments in modelling plant hydraulics (Mencuccini et al., 2019; Eller et al., 2018; Sperry et al., 2017; Baker et al., 2008) provide more accurate predictions of stomatal behaviour during drought, however, these developments must also be accompanied by models of carbon storage in order to effectively simulate the trade-off between hydraulic damage and productivity loss. Until such developments are made, predictions of plant mortality and recovery in response to climate extremes such as drought will remain uncertain.

Accurately simulating forest mortality is vital to accurate predictions of climate. This is particularly true in tropical regions where terrestrial carbon storage is large (Pan et al., 2011) and forests are frequently subjected to intense periods of environmental stress. Intense dry periods can reduce vegetation productivity and increase plant mortality in the tropics, over both short-term (Phillips et al., 2009; Bastos et al., 2018; Luo et al., 2018; Gloor et al., 2018) and multi-annual time-scales (Rowland et al., 2015; Meir et al., 2018; Metcalfe et al., 2010; Fisher et al., 2007; da Costa et al., 2010). When combined with the effects of fire and land-use change, drought can cause regions such as the Amazon basin to shift from a net sink to a net source of carbon to the atmosphere (Gatti et al., 2014; Liu et al., 2017; Phillips et al., 2009). Loss of terrestrial carbon in the Amazon represents a significant feedback loop in the climate system (Cox et al., 2000) and large losses of biomass could cause drastic and irreversible changes to the climate. However, the nature of this 'tipping point' is uncertain, and without accurate representation of forest resilience, including the balance between hydraulic failure and carbon starvation, predictions of large-scale forest die-back will remain unreliable. Drought is predicted to increase in both frequency and severity across the tropical rainforest biome in response to climate change (Marengo et al., 2018; IPCC, 2014). Accurately simulating drought responses is, therefore, a priority for the global modelling community (Corlett, 2016; Fatichi et al., 2016), although many efforts to date have focused on simulating plant hydraulic properties and have largely ignored the development of a NSC pool in models.

Despite their clear role in forest function, our current understanding of how NSCs are produced, stored and used re-

mains poor (Hartmann and Trumbore, 2016). Absolute pool sizes are difficult to quantify (Quentin et al., 2015) and it is not clear how NSC reserves are distributed and transported between different plant organs under stress (Martínez-Vilalta et al., 2016; Sevanto et al., 2014). It is also not clear whether NSC storage is the passive result of asynchrony between supply demand as described above, or whether plants also have the capacity to actively regulate NSC stores at the expense of growth and respiration (Körner, 2003; Palacio et al., 2014; Wiley and Helliker, 2012). This may go some way to explaining the apparent absence of substrate-based modelling approaches within many LSMs. Some optimised modelling studies have been conducted that explore models of NSC storage and the substrate limitation of respiration and growth (Thornley, 1970, 1971, 1972a, b, 1977, 1991, 1997, 2011; Thornley and Cannell, 2000; Dewar et al., 1999). These provide a theoretical framework to develop mechanistic models of NSC storage and utilisation (Hemming et al., 2001; Fritts et al., 2000; Salomón et al., 2019) that allow detailed simulations of plant function. However, there have been few attempts to develop such models in a manner that would be compatible with large scale LSMs (De Kauwe et al., 2014). This can largely be attributed to a scarcity of ecosystem level data (NSC content and distribution) that can be used to parametrise and evaluate models for a range of species and climates that covers all plant functional types (PFTs) used in LSMs (Fatichi et al., 2019). Site level studies that explore how the components of plant carbon expenditure respond to environmental change (e.g. Mahmud et al., 2018; Metcalfe et al., 2010) provide useful insights into the role of NSCs within a plant and can guide model development. Nonetheless, given our current knowledge and data-availability it is necessary to develop a parameter sparse model that can be calibrated against data sources that can be more effectively collected (e.g. growth and respiration data), yet capture the essential characteristics of representing a NSC pool (e.g. de-coupling photosynthesis from growth and respiration). Such an effort will not only constrain future climate projections, but may also be used to stimulate further research that improves our empirical understanding of NSC storage and use.

In this study we present 'Substrate Utilisation by Growth and Autotrophic Respiration' (SUGAR), a simplified model of substrate utilisation, designed to work within an LSM. The aim of the model is to allow the decoupling of PCE and GPP in order to provide a more accurate representation of respiration and growth fluxes, in particular in response to environmental stress. To demonstrate its behaviour and applicability to large scale ecosystem modelling, we use SUGAR to simulate PCE fluxes over the Amazon basin, using GPP data from an ensemble of LSMs, constrained by global fluorescence measurements from the Greenhouse Gases Observing SATellite (GOSAT) (Parazoo et al., 2014) as driving data. We assess the sensitivity of the model to initialised NSC content, within a reasonable range of possible pool sizes and assess the changes the model makes to predictions of ecosystem carbon expenditure. We also test the model under stressed and non-stressed conditions by simulating the world's longest running tropical rainforest through-fall exclusion (TFE) experiment and corresponding control forest in the Caxiuanã National forest, Brazil, over a 16-year period. Previous simulations of the TFE experiment by multiple LSMs has highlighted their inefficiency at capturing the effects of the artificial drought on forest function (Powell et al., 2013). It remains unclear to what extent the lack of NSC dynamics is responsible for the discrepancies between model predictions and observations in these previous studies. We examine the role NSC dynamics has on model predictions during the drought by post processing the output of one of these LSMs, namely the Joint UK Land Environment Simulator (JULES). We compare the results from JULES and the new predictions from SUGAR to observations (Metcalfe et al., 2010; da Costa et al., 2014) as well as a time-series of net primary productivity (NPP) derived from data collected in Rowland et al. (2015).

## 2 Model description

Our 'Substrate Utilisation by Growth and Autotrophic Respiration (SUGAR)' model simulates a single pool of carbohydrate at a gridbox scale, for each vegetation tile (Fig. 1). Sugars and starches are not distinguished meaning that all carbohydrate is readily available to support respiration and growth. Representing just a single pool in this way keeps the model simple and parameter sparse making integration into an LSM much easier. SUGAR is designed to sit below the photosynthesis component of a LSM. Assimilated carbon from photosynthesis (GPP) is collected by the NSC pool and the total carbon allocated to respiration and growth is then calculated and taken directly from the NSC pool. The pool is therefore always active and is constantly depleted by growth and respiration, and replenished by photosynthesis. Both growth and respiration are assumed to be single substrate enzyme reactions and depend on NSC content via the Michaeilis-Menten equation. Respiration and growth both depend on temperature via the standard $Q_{10}$ function (Ryan, 1991). Carbohydrate content is not actively regulated by the plants in SUGAR meaning that variations in NSC stores are the passive result of asynchrony between photosynthesis and PCE caused by variations in climate.

## 2.1   Non-structural carbohydrate pool

The rate of change of NSC content ($C_{NSC}$) is described by:

$$\frac{dC_{NSC}}{dt} = \Pi_G - R_p - G \tag{1}$$

where $\Pi_G$ is canopy GPP, $R_p$ is total plant respiration, and $G$ is plant growth.

Using the definition of net primary productivity ($\Pi_N$):

$$\Pi_N = \Pi_G - R_p$$

equation (1) is written as:

$$\frac{dC_{NSC}}{dt} = \Pi_N - G \tag{2}$$

To quantify the size of the NSC pool we consider an unstressed forest at steady-state. We define the average or equilibrium NSC pool size at steady state as a fraction of total structural carbon biomass and denote it by $f_{NSC}$. This is then used to initialise the NSC pool.

$$f_{NSC} = \left(\frac{C_{NSC}}{C_v}\right)^* \tag{3}$$

where $C_v$ is structural carbon biomass and the asterisk indicates steady state.

## 2.2   Growth

Plant growth depends on temperature and NSC availability. The temperature dependence is assumed to follow a $Q_{10}$ exponential relationship and the NSC dependence follows Michaelis-Menten reaction kinetics:

$$G = G_0 F_Q(T) C_v \frac{C_{NSC}}{C_{NSC} + K_m C_v} \tag{4}$$

where $G_0$ (yr$^{-1}$) is the maximum specific growth rate at the reference temperature 25°C, $T$ (°C) is temperature, $C_v$ (kg C m$^{-2}$) is total structural carbon biomass, $K_m$ is a half saturation constant equal to the NSC mass fraction at which growth rate is half of its maximum value at the reference temperature and related to the steady state NSC mass fraction by Eq. (6), and $F_Q(T)$ is the $Q_{10}$ temperature dependence given by:

$$F_Q(T) = q_{10}^{0.1(T-25)} = exp\left(ln(q_{10})\frac{(T-25)}{10}\right) \tag{5}$$

where $q_{10}$, which is a constant taken to be 2.0 by default.

The half saturation constant $K_m$ is expressed as a fraction ($a_{K_m}$) of $f_{NSC}$.

$$K_m = a_{K_m} f_{NSC} \tag{6}$$

where $a_{K_m}$ is a constant with the default value of 0.5

## 2.3   Respiration

Plant respiration is split into maintenance and growth components. Growth respiration is calculated as a constant fraction of plant growth:

$$R_g = \frac{1 - Y_g}{Y_g} G \tag{7}$$

where $Y_g$ is the growth conversion efficiency, or yield, with a default value of 0.75 (Thornley and Johnson, 1990).

Maintenance respiration has the same temperature and NSC dependence as plant growth:

$$R_m = R_{m_0} F_Q(T) C_v \frac{C_{NSC}}{C_{NSC} + K_m C_v} \tag{8}$$

where $R_{m_0}$ (yr$^{-1}$) is the maximum specific rate of maintenance respiration at the reference temperature 25°C. [5]

### 2.4   Total carbohydrate utilisation

The total rate of NSC utilisation, $U$, is defined as the sum of plant respiration and growth:

$$U = R_p + G \tag{9}$$

$U$ here is exactly equivalent to PCE and is only denoted differently for convenience and ease of reading. Using this definition, Eq. (1) can be written as: [10]

$$\frac{dC_{NSC}}{dt} = \Pi_G - U \tag{10}$$

Since both respiration and growth have the same NSC and temperature dependence, $U$ is given by:

$$U = \phi F_Q(T) C_v \frac{C_{NSC}}{C_{NSC} + K_m C_v} \tag{11}$$

where $\phi = R_{m_0} + \frac{G_0}{Y_g}$ is the maximum specific rate of utilisation of carbohydrate at the reference temperature 25°C.

[15]

### 3   Parameter estimation

Detailed time-series data of forest level NSC stocks are extremely difficult to collect and are therefore scarce. This makes parameter evaluation difficult. Here we discuss the evaluation process for each parameter in SUGAR. Some of these parameters, for example the $q_{10}$ parameter, have standard or commonly used values within the LSM literature, and the validity of their given values is beyond the scope of this paper. In these cases we present only a very brief justification. For the remaining [20] parameters, we outline how with a few assumptions we can use the simplicity of SUGAR to evaluate these parameters without the need for detailed NSC data, and instead use more commonly and readily measured variables. An overview of all parameters, their default values and the values used in this study is given in table (2).

  – $q_{10}$ [25]
    The $q_{10}$ parameter represents the factor by which respiration and growth increase with every 10°C of warming. The exponential $Q_{10}$ function is commonly used to describe the temperature dependence of plant metabolism in LSMs with the standard $q_{10}$ value of 2.0 (Ryan, 1991)

  – $Y_g$
    The $Y_g$ parameter represents the conversion efficiency of plant growth (Thornley and Johnson, 1990). By default, we [30] assume a value of 0.75, consistent with previous estimates (Thornley and Johnson, 1990), and the parameters assumed in other LSMs (e.g. Clark et al., 2011) It is derived in Thornley and Johnson (1990) and estimated to equal 0.75. Similar parameters are used in other LSMs (e.g. Clark et al., 2011).

  – $f_{NSC}$
    The $f_{NSC}$ parameter represents the non-stressed, equilibrium NSC pool size as a fraction of total structural carbon. This [35] can be set directly using empirical data (e.g. for tropical forests using Würth et al. (2005)). Note that studies such as Würth et al. (2005) present NSC stocks as a fraction of total dry mass and so data should be adjusted to account for non-carbon mass.

– $\phi$

The $\phi$ parameter represents the maximum specific rate of carbohydrate utilisation by plant respiration and growth at the reference temperature of 25°C. To estimate $\phi$ we consider an unstressed forest at steady-state with an NSC fraction of $f_{NSC}$. Under these circumstances the forest is neither significantly drawing upon nor adding to the NSC stores and the turnover rate of NSC must equal to the carbon assimilated by photosynthesis. This allows the following expression for $\phi$ to be found in terms of GPP and forest biomass which are more easily measured at an ecosystem scale than total NSC stocks:

$$\phi = \frac{1 + a_{K_m}}{F_Q^*(T)} \left( \frac{\Pi_G}{C_v} \right)^* \tag{12}$$

Where the asterisk denotes a temporal average over the period $\tau_{obs}$. i.e for variable $X$:

$$X^* = \frac{1}{\tau_{obs}} \int\limits_{\tau_{obs}} X dt \tag{13}$$

To evaluate $\phi$, we require an estimate of average specific GPP and average temperature over the period of observation. If SUGAR is used at a single site these can be evaluated directly using GPP, biomass and temperature data where these are available. If these data are not available then the specific GPP can be approximated as the steady state carbon residency time, $\tau = \frac{C_v}{\Pi_G}$ (e.g. Carvalhais et al., 2014), and the temperature can found using global climatology data over the same period.

– $G_0$ and $R_{m_0}$

These parameters represent the maximum specific rate of plant growth and maintenance respiration respectively, at the reference temperature of 25°C. To evaluate these parameters we define the parameter $\alpha$ as the ratio of $G_0$ to $\phi$:

$$\alpha = \frac{G_0}{\phi} \tag{14}$$

We can then evaluate $\alpha$ by again considering a non-stressed forest in steady state, when it is equal to the average carbon use efficiency (CUE) over the period of observation:

$$\alpha = CUE^* \tag{15}$$

Again this can be evaluated using data from a single site where available, or using more general estimates of CUE (e.g. Chambers et al., 2004; Gifford, 1995) if not.

We then can find $G_0$ and $R_{m_0}$ as:

$$G_0 = \alpha\phi \tag{16}$$

and

$$R_{m_0} = \left( 1 - \frac{\alpha}{Y_g} \right) \phi \tag{17}$$

– $a_{K_m}$

The $a_{K_m}$ parameter relates the half saturation constant ($K_m$) to the equilibrium NSC pool size ($f_{NSC}$). It is currently not possible to evaluate $a_{K_m}$ from empirical data. We give this parameter a value of 0.5, as this gives realistic NSC mass fractions. The sensitivity of the SUGAR model to this parameter is examined in this study within the range $a_{K_m} \in [0.1, 2.0]$.

## 4   Methods

### 4.1   Sensitivity study over the Amazon-Basin

To demonstrate how SUGAR influences predictions of PCE, we conduct a series of simulations over a six and a half year period from June 2009 to December 2015, across the whole Amazon, where $f_{NSC}$ is varied from 0.0005-0.16. As $f_{NSC}$

represents the initial fraction of the biomass pool that is NSC, a value of 0.0005 is effectively representing a model without NSC. The upper bound of 0.16 is an estimate of the ecosystem NSC content in a tropical forest in Panama (Würth et al., 2005). The model is driven with monthly GPP data from an ensemble of LSMs constrained by global fluorescence measurements from the Greenhouse Gases Observing SATellite (GOSAT) (Parazoo et al., 2014), and temperature data from CRU-JRA (Harris, 2019). SUGAR is parametrised as described above with parameters $Y_g$, $a_{K_m}$ and $q_{10}$ kept at their default values. A value for $\phi$ is found for each grid-box using biomass estimates across the Amazon (Avitabile et al., 2016) and the first year of GOSAT GPP.

To assess the effect that the SUGAR model has on predictions of PCE, a basin wide average PCE flux is compared to the basin average GPP for each value of $f_{NSC}$. The Pearson correlation coefficient of simulated PCE and driving GPP, and PCE and the $Q_{10}$ function in each grid cell is also calculated for each value of $f_{NSC}$ and presented on maps.

## 4.2 Methods - Simulating responses to drought

To evaluate the effectiveness of SUGAR at simulating responses to drought, we tested it at the world's longest tropical drought experiment.

### 4.2.1 Site Description

The TFE experiment is located in Caxiuanã National Forest, Pará State, Brazil (1°43'3.5"S, 51°27'36"W), where measurements of meteorology and plant physiology of two 1ha plots began in 2001. In January 2002, panels were introduced into one of the plots, excluding c. 50% of rainfall from the soils and subjecting the plot to an artificial drought. Measurements of meteorology and forest physiology continue to the present day (this study looks only up to 2016-12-09). During this period mean annual rainfall was between 1772.6 and 2967.1 mm. Daily incident radiation varied from 419.8 $\mathrm{Wm}^{-2}$ to 731.1 $\mathrm{Wm}^{-2}$. A full summary of experimental set up and the most recent collection of results from the site is available in Meir et al. (2018).

At the start of the experiment, total estimated above-ground biomass was 213.9±14.2 $\mathrm{Mgha}^{-1}$ in the control forest, and 200.6±13.2 $\mathrm{Mgha}^{-1}$ in the TFE plot. After 13 years of the drought treatment, biomass loss to mortality in the TFE plot had increased by 41.0±2.7% relative to 2001 values (Rowland et al., 2015). Observations and modelling studies at the site suggest that while GPP declined in response to the artificial drought, PCE was maintained at close to pre-drought levels during at least the first 3-4 years of the experiment (Metcalfe et al., 2010; Fisher et al., 2007). NSC reserves are thought to have sustained PCE during this time and it is estimated that the forest had access to c. 20 $\mathrm{MgCha}^{-1}$ of available NSC (c. 8% of live biomass) during the drought (Metcalfe et al., 2010). It is not possible for LSMs to accurately predict both growth and respiration in the TFE forest without simulating some kind of NSC storage, and makes the experiment an ideal opportunity to test SUGAR.

### 4.2.2 Simulation descriptions

The TFE experiment and corresponding control plot are simulated over the period 2001-01-01 to 2016-12-09. The first set of simulations are conducted using the Joint UK Land Environment Simulator (JULES) (Best et al., 2011; Clark et al., 2011), driven with the meteorological data collected at Caxiuanã. JULES version 5.2 is used with a pre-existing parametrisation of the site and then optimised so that annual GPP and NPP in the control forest agree with observations. The same configuration is then used to simulate the TFE forest. Both control and TFE plot were initialised and spun up for 176 years using a repeated loop of the control meteorological data. To simulate the effect of the drought experiment, precipitation is halved in the TFE simulation from January 1, 2002, in line with estimates of average exclusion rate.

Gridbox GPP (gpp_gb) and grid-box temperature at 1.5 m above canopy height (t1p5m_gb) outputs from JULES are then used to drive the SUGAR model off-line in each plot. In order to examine how SUGAR compares relative to JULES, it is parametrised using the first year of output data from JULES (i.e. the year before panels are put in the TFE plot) rather than observations from Caxiuanã (with the exception of an estimate of NSC pool size ($f_{NSC}$), which is necessary given JULES does not model NSC). The average GPP and biomass of the simulated forest is used to find average specific GPP which is used to evaluate $\phi$. The parameter $\alpha$ is evaluated by finding the average CUE of the simulated forest over this year which is then used to evaluate $R_{m_0}$ and $G_0$. Since the SUGAR simulations are off-line (i.e. not coupled to a Dynamic Global Vegetation Model (DGVM)) we assume that biomass ($C_v$) remains constant throughout the experiment. This is a necessary assumption that allows the simulations to be performed off-line and the effect of the NSC pool to be examined in isolation. Finally, to test the sensitivity of SUGAR to the parameter $a_{K_m}$, it is varied from 0.1 to 2.0.

### 4.2.3   Model Evaluation

Snapshot fluxes (NPP, $R_p$, PCE) from JULES and SUGAR are evaluated against observations from (Metcalfe et al., 2010) and (da Costa et al., 2014) for the periods 2005 and 2009-2011. Model growth output is evaluated against an observed time-series of NPP from both plots. Observed NPP does not include root increment due to the difficulty in measuring total root growth at the plot level scale. It is therefore calculated using the above-ground biomass (AGB) increment and total local litter-fall (Rowland et al., 2018). Both model outputs are altered by removing simulated root increment. In SUGAR this is carried-out using the allometric scaling within JULES. Biomass increment is calculated using tree trunk diameter at breast height (DBH) data and a number of allometric equations (Table 3. The DBH data were collected every 1-3 years for each tree in each plot using dendrometers between July 2000 and December 2014(Rowland et al., 2015). The error bars presented are the sum of measurement error from the litterfall data and the 95% confidence intervals of the ensemble of allometric equations. Scaling NSC measurements to a whole plant and whole plot scale is difficult and has large associated errors (Quentin et al., 2015). We therefore evaluate SUGAR primarily against integrated flux and biomass increment data.

## 5   Results

### 5.1   Sensitivity study over the Amazon-Basin

In simulations of PCE across the Amazon Basin, the SUGAR model dampens the seasonal variations in both respiration and growth, relative to GPP, maintaining a less variable rate of PCE (Fig. 2). We present the coefficient of variation (CV) of the basin averaged GPP and simulated PCE for each value of $f_{NSC}$. We also present the grid-box bounds of CV which is the coefficient of variation of the least and most variable grid-boxes for each simulation. The CV of the basin average GPP data is 9.51% (grid-box bounds: 7.47 – 40.9%, Fig. A1). When the SUGAR model is initialised with $f_{NSC}$ = 0.0005, effectively representing a model with no NSC, the CV of the basin averaged PCE is 9.12% (grid-box bounds: 6.57 – 37.4%, Fig. A1). As $f_{NSC}$ increases the coefficient of variation decreases sharply across all grid boxes. At $f_{NSC}$ = 0.04, the CV of variation across the Amazon is 3.73% (grid-box bounds: 3.59 - 29.8%, Fig. A1). The dampening effect starts to saturate at larger values of $f_{NSC}$ and the CV of simulated PCE decreases more slowly with increasing $f_{NSC}$ from this point. At $f_{NSC}$ = 0.08, the CV of PCE across the Amazon is 3.54% (bounds: 3.78 – 25.1%, Fig. A1). Finally at $f_{NSC}$ = 0.16 the CV of simulated basin PCE is 3.63% (grid-box bounds: 3.74 - 22.9%, Fig. A1). Increasing the effective size of the NSC pool also reduces the spatial variation in PCE seasonality across Amazonia. Relative to the wetter northern Amazon, the more seasonally dry southern Amazon experiences far greater seasonal variation in GPP. This pattern is mirrored in the seasonal variation of simulated PCE, however, with more NSC in the model the difference between PCE seasonality in the north and south declines, due to a larger decrease in seasonal variation of growth and respiration in the southern regions. This decline in seasonal variation is caused by an increase in dry season carbon expenditure and a decrease in the wet season carbon expenditure. The buffering effect is a consequence of the de-coupling of respiration and growth from GPP, reflected in the decline in the mean correlation coefficient between GPP and PCE from 0.980 (bounds: 0.939 – 1.00) to 0.181 (bounds -0.501 – 0.997) from simulations with the 0 to 8% mass fraction of NSC (Fig. 3). With this decoupling effect there is also a shift in the primary driver of simulated PCE, from GPP (in the 0% NSC mass fraction simulation) to the $Q_{10}$ function (in the 8% NSC mass fraction simulation). This is reflected in the increase in the mean correlation coefficient between simulated PCE and the $Q_{10}$ function (Eq. (5)) in SUGAR from -0.0485 (bounds: -0.651 to 0.517) to 0.637 (bounds: -0.456 to 0.956) in the 0 to 8% NSC mass fraction simulations (Fig. 4).

### 5.2   Simulations in a tropical moist forest

In the simulations of the control plot, in which the forest was not subject to any artificial drought stress, JULES and SUGAR produce similar results of long term NPP accumulation (Fig. 5), that are both consistent with observations. By the end of the NPP observation period (2014-12-17), JULES predicts a total accumulated NPP of 155.6 MgCha$^{-1}$ and SUGAR 154.7 MgCha$^{-1}$. Both results are consistent with observations (Fig. 5, 161.5±22.0 MgCha$^{-1}$) from the site.

There are some larger differences between JULES and SUGAR on annual time-scales, but in general the models predict comparable annual mean values of control plot PCE, Ra and NPP (Fig. 6). During the first three years of the experiment (2002, 2003, 2004), JULES predicts an annual mean PCE of 35.13 MgCha$^{-1}$yr$^{-1}$, and SUGAR predicts 34.79±0.17 MgCha$^{-1}$yr$^{-1}$. Both these results lie within the confidence intervals of the observations from the site (Fig. 6, 33.0±2.9 MgCha$^{-1}$yr$^{-1}$). The two models differ most in the natural drought years of 2005, 2010 and 2015 in which predicted annual GPP is at its lowest. In 2005 JULES predicts a decrease (relative to the 2002-2004 period) in annual mean PCE to 33.32

$\text{MgCha}^{-1}\text{yr}^{-1}$ (-5.15%) whereas SUGAR predicts an increase to $36.13\pm0.27$ $\text{MgCha}^{-1}\text{yr}^{-1}$ (+3.85%). The decrease in JULES PCE is caused by a decrease in predicted GPP in 2005. In SUGAR this decrease in GPP is buffered by NSC storage (Fig. 7), and increase in the annual mean temperature drives the increase in predicted PCE. Both results are close to the observed value although the SUGAR result is outside the observed confidence intervals by 0.64%. In 2010 average annual rainfall was 1772.6 $\text{mmyr}^{-1}$, the lowest in the 16 year period (c. 25% decrease on the 16-year mean 2324.2 $\text{mmyr}^{-1}$). This causes a decline in predicted GPP on the control plot from 35.92 $\text{MgCha}^{-1}\text{yr}^{-1}$ in 2008 to 32.94 $\text{MgCha}^{-1}\text{yr}^{-1}$ in 2010. Consequently, JULES predicts a mean PCE of 33.60 $\text{MgCha}^{-1}\text{yr}^{-1}$ over the period 2009-2011 which lies below observed values. SUGAR is able to buffer the forest against the 2010 decline in GPP and allows elevated PCE in 2010 ($36.36\pm0.36$ $\text{MgCha}^{-1}\text{yr}^{-1}$) relative to 2008 ($34.52\pm0.52$ $\text{MgCha}^{-1}\text{yr}^{-1}$). This allows SUGAR to maintain a mean PCE value over the 2009-2011 period of $36.00\pm0.54$ $\text{MgCha}^{-1}\text{yr}^{-1}$ which is close to observations (Fig. 6).

## 5.3 Simulating responses to drought

In the TFE plot simulations, SUGAR and JULES diverge significantly in their predictions of NPP, PCE and Ra, with SUGAR more accurately capturing observations than JULES (Figs. 5&6). JULES is able to capture NPP accumulation for approximately 1 year after the start of the drought treatment, however, from 2003 onwards, predicted NPP accumulation drops significantly below the confidence intervals of the observations (Fig. 5). This is driven predominantly by a sharp decline in GPP in response to the declining water availability. SUGAR is able to capture NPP accumulation for much longer and predictions remain within the confidence intervals of the observations until the start of 2009 (Fig. 5). By the end of the observation period JULES predicts a total of 60.6 $\text{MgCha}^{-1}$ of accumulated and SUGAR 105.22 $\text{MgCha}^{-1}$. Neither result lies within observed confidence intervals of the observations (Fig. 5, $126.8\pm16.9$ $\text{MgCha}^{-1}$) although the SUGAR result represents a significant improvement relative to JULES.

During the first 3 years of the experiment, SUGAR is able to buffer a significant decline in predicted GPP on the TFE plot, which drops from 34.90 $\text{MgCha}^{-1}\text{yr}^{-1}$ in 2001, to a minimum of 19.61 $\text{MgCha}^{-1}\text{yr}^{-1}$ in 2003 (-43.8%). Since JULES does not contain an NSC storage component and PCE is equal to GPP, PCE in JULES also drops by 43.8%, from 34.90 $\text{MgCha}^{-1}\text{yr}^{-1}$ in 2001 to 19.61 $\text{MgCha}^{-1}\text{yr}^{-1}$ in 2003. As a result JULES predicts a mean PCE value of 24.84 $\text{MgCha}^{-1}\text{yr}^{-1}$ over the first three years of drought treatment (2002, 2003, 2004). These values are outside the confidence intervals of the observations and 26.7% below the mean PCE value observed in the TFE plot ($33.9\pm3.6$ $\text{MgCha}^{-1}\text{yr}^{-1}$, Fig. 6). The SUGAR model is able to maintain PCE at a higher level than JULES during these first three years by drawing upon a mean $5.60\pm1.01$ $\text{MgCha}^{-1}$ of NSC each year to support growth and respiration (Fig. 7). This results in a mean PCE of $30.44\pm1.01$ $\text{MgCha}^{-1}\text{yr}^{-1}$ over the period 2002-2004, which lies within the observed confidence interval (Fig. 6). The NSC buffering effect in SUGAR continues in 2005 with SUGAR expending $5.59\pm0.76$ $\text{MgCha}^{-1}$ more carbon than JULES during that year. This means that the predicted annual mean PCE in SUGAR is $22.82\pm0.76$ $\text{MgCha}^{-1}\text{yr}^{-1}$ compared to 17.23 $\text{MgCha}^{-1}\text{yr}^{-1}$ in JULES. Both results lie below the lower bound of the observed confidence intervals ($33.9\pm3.6$ $\text{MgCha}^{-1}\text{yr}^{-1}$, Fig. 6), however, the SUGAR result represents a significant improvement relative to JULES. In the latter years of the drought simulations (2009 onwards), the NSC pool becomes significantly depleted (Fig. 7) and the buffering effect in SUGAR (described above) diminishes. Consequently, on annual time-scales, the mean PCE in JULES and SUGAR during the 2009-2011 period are similar (20.76 and $21.20pm0.87$ $\text{MgCha}^{-1}\text{yr}^{-1}$ respectively), although the allocation of carbon to respiration and growth is different, with SUGAR expending more ($6.70\pm0.28$ $\text{MgCha}^{-1}\text{yr}^{-1}$) carbon on growth than JULES (3.06 $\text{MgCha}^{-1}\text{yr}^{-1}$). This difference in allocation allows SUGAR to predict the observed NPP with more skill than JULES, however it means that respiration predictions are reduced relative to JULES and the observations.

## 6 Discussion

SUGAR alters the relationship between photosynthesis and carbon expenditure. This has implications for simulations of both extreme and more gradual changes in climatic and meteorological conditions. By decoupling PCE from GPP, SUGAR creates a buffering effect that decreases the seasonal variation of carbon expenditure, even in ecosystems where the variation of GPP is already low. As we increase the levels of stored substrate within our simulations, the variability in PCE declines, due to an increased ability to maintain respiration and growth when GPP is low, and replenishment of the NSC pool when GPP is high. This effect is most pronounced in the semi-arid regions of the southern Amazon where there is a strong seasonal cycle in GPP (Fig. A2), corresponding to a strong seasonal pattern of precipitation. Semi-arid regions provide the largest contribution to the global carbon sink anomaly, in part due to this high variability in GPP (Poulter et al., 2014; Ahlström et al., 2015). To represent this contribution, land surface models must capture the response of vegetation to the climate variability experienced

in these regions now and in the future. SUGAR provides a mechanistic approach to achieve this by simulating respiration and growth as a separate function to GPP. Given the strong evidence from observations that NPP and respiration do not have the same seasonal and climatic responses as GPP (Liu et al., 2017; Girardin et al., 2016; Doughty et al., 2015a), accurately predicting future variability in atmospheric $CO_2$ concentrations (Cox et al., 2013) will be reliant on a sub-model such as SUGAR which can allow this de-coupling to occur. Research demonstrating the importance of highly seasonal arid regions highlights the necessity of substrate-based approaches in large scale ecosystem models and should motivate the community to focus on improving our understanding of NSCs and how to model them.

The sensitivity of the biosphere to climate change has large impacts on the future climate. For example, large losses of tropical forest carbon may represent a tipping point in the climate system that could have highly adverse and irreversible consequences for the global climate (Cox et al., 2000). However, both the nature and likelihood of such a tipping point is uncertain. Feedbacks between the climate and the carbon cycle mean that small perturbations in the state of the biosphere can make significant changes to the future state of the climate (Friedlingstein et al., 2001). Small changes in the sensitivity of a tropical forest to climate change, may be the difference between the continued absorption of $CO_2$ by ecosystems such as the Amazon, and the severe die-back scenarios predicted by some models (Huntingford et al., 2013; Phillips et al., 2009). Therefore the difference between a forest that is able to buffer the effects of even a short drought or reduction in productivity, and a forest that is not, may be significant at a global context in the future, even if it appears small in the present day. Non-conservative propagation of perturbations in the state of vegetated ecosystems contributes to large uncertainty in climate models (Huntingford et al., 2009), which greatly reduces our ability to constrain future climate possibilities and tipping points within the carbon-cycle. Accurately representing the response of forest biomass, particularly in the tropics, to changes in climate is crucial to reducing this uncertainty and is a major goal of the climate and land surface modelling community. The buffering effect demonstrated in SUGAR may have an indirect yet large impact on the predictions of future climate by LSMs and provide a more realistic representation of forest sensitivity to climate.

As well as a buffering of carbon expenditure, SUGAR also enables a transition of the primary driver of growth and respiration. With little or no carbohydrate, carbon expenditure in SUGAR is driven predominantly by the rate of photosynthesis (Fig 4). Carbon is used by the ecosystem as soon as it is assimilated, meaning that the rate of expenditure is highly correlated with the rate of photosynthesis. This is often described as 'source driven carbon dynamics' meaning that photosynthesis is the key driving flux in determining the carbon balance of the ecosystem. 'Source driven carbon dynamics' are at the centre of many LSMs including JULES. As more carbohydrate is added to the ecosystem in SUGAR, temperature becomes the predominant driver of PCE via the $Q_{10}$ function (Eq. (5), Fig. 4). As more carbon is stored, growth and respiration become less carbon limited and more controlled by the $Q_{10}$ function within SUGAR. This shift can be seen as a transition towards 'sink driven carbon dynamics'. Under the theory of sink driven carbon dynamics, environmental variables such as temperature and water-availability exert a direct control over carbon expenditure that can be larger than that of photosynthesis (Körner, 2003; Wiley and Helliker, 2012; Palacio et al., 2014; Fatichi et al., 2014). Processes such as end-product inhibition (Stitt, 1991), in which photosynthesis is inhibited by an excess of assimilate in the leaves, mean that growth and respiration may even exert indirect control over the rate of photosynthesis. The result is that 'sink' fluxes (i.e respiration and growth), driven by environmental variables, are the predominant determinants of ecosystem carbon balance. Since the NSC pool in SUGAR does not exert any control over photosynthesis (e.g. via end-product inhibition) the behaviour of SUGAR here cannot be described as truly sink driven. However, SUGAR provides a framework that allows processes such as end-product inhibition to be implemented, and so provides the opportunity to represent both sink and source driven dynamics in LSMs. This allows a greater representation of how the limiting factors of growth and respiration interact with, and respond to a changing climate.

Using the Caxiuanã control simulations we demonstrate that SUGAR and JULES predict very similar long-term NPP accumulation in the natural climate conditions of a tropical moist forest. However, there are larger differences between SUGAR and JULES on an annual time-scale, due to the buffering of the natural variability in GPP by SUGAR. These results further highlight the importance of substrate-based modelling to better capture the responses to natural variation, even under current climate conditions and without extreme events (Doughty et al., 2015a). In the TFE plot, SUGAR makes significant improvements to the prediction of ecosystem carbon fluxes, particularly for accumulated NPP. This improvement is caused by a combination of two processes that occur in SUGAR and that are not present in JULES. The first process is the utilisation of the NSC pool during the early stages of the experiment. SUGAR expends a mean 5.53 $\mathrm{Mgha^{-1}}$ more carbon than is assimilated through photosynthesis in the first three years of drought (2002-2004) and a further 5.80 $\mathrm{Mgha^{-1}}$ in 2005. This allows an increase in both NPP and respiration relative to JULES and is consistent with the analysis in Metcalfe et al. (2010), which suggests the TFE plot was expending $7\pm4.5$ $\mathrm{MgCha^{-1}yr^{-1}}$ more than it was accumulating in 2005, implying that NSC stores were being depleted in response to the drought. The second process is the down regulation of respiration in response to

the depleting NSC pool. In the JULES simulations, photosynthesis declines much faster than respiration and, since growth is equal to GPP – Ra in JULES, this means that NPP drops significantly as GPP declines in response to the drought. The result of this effect is that in two years (2005 and 2007), the predicted annual mean NPP by JULES, is negative. Negative NPP is generally considered to be unrealistic, particularly over the time-scale of a year (Roxburgh et al., 2005), and since JULES does not contain a labile carbon pool to support the deficit, missing carbon is taken from the structural pool. The physical 5 interpretation of this is that trees in JULES respire away their structural carbon and shrink. While there is some evidence of recycling and remobilisation of structural compounds, the magnitude of structural carbon being allocated to respiration (via the resulting negative NPP) in these JULES simulations is not realistic. In SUGAR, respiration declines due to the depletion of the NSC pool. This down-regulation of Ra means that a larger proportion of instantaneous GPP is available for NPP, resulting in larger predictions of NPP in SUGAR than JULES, despite similar estimates of total PCE. While this latter process aids the 10 prediction of NPP in SUGAR, it should be noted that observations from Caxiuanã actually indicate an increase in TFE plot respiration between 2005 and 2011 (Metcalfe et al., 2010; da Costa et al., 2014). SUGAR is currently unable to capture this increase and this is likely due to the simplicity of the assumptions made within the model. For example, we have assumed that plant growth is directly dependent on carbohydrate availability and temperature only. Water stress may reduce plant growth in SUGAR, but only indirectly by inhibiting photosynthesis and causing a decrease in available carbon. However, in reality 15 plant growth can be affected directly by decreasing water availability through the inhibition of cambial expansion (Balducci et al., 2013; Hsiao, 1973; Boyer, 1970). This decline in growth may even occur before declines in photosynthesis which can cause a build up of NSC and eventually result in an increase in respiration (Fatichi et al., 2014). We are not suggesting that this specific process explains the observed increases in respiration on the drought plot at Caxiuanã, but such interactions between NSC utilisation and the environment are likely to have been important during the TFE experiment. Neither SUGAR 20 nor JULES are able to capture these processes currently. However, by implementing SUGAR within JULES we create a basis upon which we can start to represent these interactions and continue to improve predictions of forest responses to drought.

The ability of SUGAR to accurately capture PCE responses to drought in these simulations is also somewhat limited by the GPP used to run it. Photosynthesis in JULES has a high sensitivity to reductions in soil moisture (eg., Harper 25 et al., 2016; Williams et al., 2018). In the Caxiuanã simulations JULES predicts an average decline in annual GPP of 4.42 $MgCha^{-1}yr^{-1}$ from 2001 to 2005 in the TFE plot. Combining the observed PCE rates in the TFE plot with the predicted GPP by JULES would imply that the forest is using an average of 10.96 $MgCha^{-1}yr^{-1}$ carbon more than it is assimilating in the first four years. This would then imply that the forest has access to at least 43.86MgC/ha of NSC, c. 22% of estimated forest biomass. Such a high NSC content is unlikely for tropical forests, which are more likely to have reserves close to 10% (Würth 30 et al., 2005). The other, and more likely explanation is that JULES is overestimating the decline in photosynthesis in response to the drought. To test this, we artificially reduced drought stress in JULES by 50% and repeated the Caxiuanã simulations (Figs. A5 & A6). This improved predictions of PCE in both models, supporting the hypothesis that JULES overestimates the sensitivity of photosynthesis to drought at this site. The recent work to improve stomatal responses to drought stress (Mencuccini et al., 2019; Eller et al., 2018; Sperry et al., 2017) has the potential to significantly improve GPP predictions in 35 LSMs such as JULES. However, there is a clear link between hydraulics and labile carbon storage, given stomatal closure comes at the cost of a reduction in carbon assimilation. The ability of a plant to store and use labile carbon is crucial to its ability to survive, and recover from, drought-induced stomatal closure (Sala and Mencuccini, 2014; O'Brien et al., 2014; Trugman et al., 2018). Without including at least simple representations of NSC storage, the potential of this recent work to improve the representation of stomatal behaviour in response to drought in LSMs, is unlikely to be realised. 40

SUGAR is a purposefully simple model of NSC storage and is missing some key processes known to be important in defining the complexities of NSC storage and use within a plant. A more complex NSC model might, for example, distinguish between starch and sugar pools, or represent multiple pools for each plant organ, and actively control the input or output of NSC into pools (Martínez-Vilalta et al., 2016; Hartmann and Trumbore, 2016). However, such models would likely require 45 representation of substrate transport between pools and the scaling of NSC data to the level of trees and forests. Recent advancements in measurement protocols may allow these datasets to be reliably collected (Landhäusser et al., 2018), however, previously the level of uncertainty on such figures has been up to 400% (Quentin et al., 2015). As a result, comprehensive NSC data-sets, measured through time in response to climatic variations and across enough biomes to allow all model PFTs to be evaluated are currently not available. Therefore, this is not a currently viable way to constrain model output. SUGAR is 50 designed to break the direct link between PCE and GPP found in many LSMs and to provide more mechanistic predictions of growth and respiration. It can be parametrised, initialised and evaluated with data that is commonly collected across the globe – Biomass, GPP and temperature; CUE (to find $\alpha$); and respiration and NPP (for evaluation). It also requires an input of initialised NSC fraction ($f_{NSC}$) which is not easily measured for an ecosystem, although values of $f_{NSC}$ can be constrained within sensible bounds (Würth et al., 2005). It may also be possible to use SUGAR as a tool to further constrain observed 55

values of NSC content by conducting sensitivity studies of $f_{NSC}$. Given the existing level of knowledge, it is more robust and realistic to use a simple model such as SUGAR which can be evaluated against more easily available observations such as Ra, PCE, NPP and GPP. As the accuracy and spatial extent of NSC data grows models such as SUGAR can act as a simple skeleton that allows new processes to be implemented into LSMs, to more accurately represent the complexity of plant carbon
storage and use.

## 7    Conclusions

We have developed a simple model of NSC storage, designed to be integrated into an LSM. The model makes significant changes to the variability of growth and respiration predictions in both extreme and more stable climatic conditions. This has large implications for simulations of future climate given the importance of predicting the variability of atmospheric $CO_2$ con-
10 centrations. The model also allows a more mechanistic representation of the limiting factors of carbon expenditure which may become increasingly important as the climate changes in the future. Due to the simplicity of the model it is easily parametrised using pre-existing data and does not require complex datasets of NSC storage which are currently unavailable. This makes the model attractive since it can be easily integrated into LSMs without introducing unreasonable uncertainty in parameter values. The magnitude of the change demonstrates the importance of representing carbon storage in LSMs and we hope will
motivate both the modelling and empirical communities to further develop our understanding and model representation of NSC dynamics.

*Code availability.*  A model example of SUGAR for a single site and set up to run at Caxiuanã using output from JULES is available at http://doi.org/10.5281/zenodo.3547613 For further information or code please contact sj326@exeter.ac.uk

## Appendix A:  Derivation of model parameters

**A1    Derivation of $\phi$**

We start by finding the rate of change of NSC mass fraction, $W_{NSC} = \dfrac{C_{NSC}}{C_v}$, in terms of $C_{NSC}$ and $C_v$:

$$\frac{dW_{NSC}}{dt} = \frac{1}{C_v}\frac{dC_{NSC}}{dt} - W_{NSC}\frac{1}{C_v}\frac{dC_v}{dt} \tag{A1}$$

We consider the case where the NSC mass fraction is constant and the left hand side of equation (A1) is zero. In reality the
25 NSC mass fraction of forest will not be exactly constant and variations in environmental variables will cause changes in NSC stocks. However, for a non-stressed forest it is a good assumption that over a prolonged period, $\tau_{obs}$, the NSC mass fraction will be roughly constant. For example, we can assume that over the course of one year, a non-stressed forest will use as much carbon as it assimilates and consequently will end the year with roughly the same NSC stock with which it started. This means that we can integrate equation (A1) over this period and set the left hand side equal to zero:

$$0 = \int_{\tau_{obs}} \left( \frac{1}{C_v}\frac{dC_{NSC}}{dt} - W_{NSC}\frac{1}{C_v}\frac{dC_v}{dt} \right) dt \tag{A2}$$

Since we are considering a forest in steady-state, we can neglect the rate of change of structural biomass, $\dfrac{dC_v}{dt}$

$$\frac{dW_{NSC}}{dt} = \frac{1}{C_v}\frac{dC_{NSC}}{dt} \tag{A3}$$

We then use the equation 1 for the rate of change of NSC:

$$0 = \int_{\tau_{obs}} \left( \frac{\Pi_G}{C_v} - \frac{R_p}{C_v} - \frac{G}{C_v} \right) dt \tag{A4}$$

To evaluate $\phi$ we use the equation for total carbohydrate utilisation and rearrange:

$$\phi \int_{\tau_{obs}} F_Q(T) \frac{W_{NSC}}{W_{NSC} + K_m} dt = \int_{\tau_{obs}} \frac{\Pi_G}{C_v} dt \tag{A5}$$

We divide both sides by $\tau_{obs}$ and assume that this can be approximated as:

$$\phi F_Q^*(T) \frac{W_{NSC}^*}{W_{NSC}^* + K_m} = \left(\frac{\Pi_G}{C_v}\right)^* \tag{A6}$$

Where the asterisk denotes a temporal average over the period $\tau_{obs}$. i.e for variable $X$:

$$X^* = \frac{1}{\tau_{obs}} \int_{\tau_{obs}} X dt \tag{A7}$$

Rearranging, we find the expression for $\phi$

$$\phi = \frac{W_{NSC}^* + K_m}{F_Q^*(T) W_{NSC}^*} \left(\frac{\Pi_G}{C_v}\right)^* \tag{A8}$$

By definition, the average NSC mass fraction is equal to $f_{NSC}$. Using this and equation (6), this becomes

$$\phi = \frac{1 + a_{K_m}}{F_Q^*(T)} \left(\frac{\Pi_G}{C_v}\right)^* \tag{A9}$$

This means that to evaluate $\phi$, we require an estimate of average specific GPP and average temperature over some reasonable stable unstressed period. If SUGAR is used at a single site these can be evaluated directly using GPP, biomass and temperature data where available. If these data are not available then the specific GPP can be approximated as the steady state carbon residency time, $\tau$ (e.g. Carvalhais et al., 2014) and the temperature can found using global climatology data over the same period.

## A2 Derivation of $\alpha$

We re-write equation (A4) as:

$$0 = \int \left(\frac{\Pi_N}{C_v} - \frac{G}{C_v}\right) dt \tag{A10}$$

Again we divide by the integration period, $\tau_{obs}$, and assume this can be written as:

$$0 = \frac{\Pi_N^*}{C_v^*} - \frac{G^*}{C_v^*} \tag{A11}$$

hence:

$$\Pi_N^* = G^* \tag{A12}$$

Similarly using equation (A4), we find

$$\Pi_G^* = U^* \tag{A13}$$

Dividing equation (A12) by equation (A13) gives:

$$\alpha = CUE^* \tag{A14}$$

where $CUE^* = \dfrac{\Pi_N^*}{\Pi_G^*}$, is the time averaged carbon use efficiency of the non-stressed forest over the period $\tau_{obs}$.

*Author contributions.* SJ, LR, PC, DH and AH developed the SUGAR model code and SJ carried out the simulations. JULES simulations were conducted by SJ with contributions from AH and KW. GPP data from GOSAT were provided by NP. Data from the through-fall exclusion experiment in Caxiuanã were provided by AdC, PM and LR. SJ prepared the manuscript with contributions from all co-authors.

*Competing interests.* The authors declare that they have no conflict of interest

*Acknowledgements.* This work has been funded by the Natural Environmental Research Council NE/L002434/1. LR was supported by the UK NERC independent fellowship grant NE/N014022/1. KW was supported by the Newton Fund through the Met Office Climate Science for Service Partnership Brazil (CSSP Brazil). PM was supported by NERC NE/N006852/1 and ARC DP170104091. MM was supported by the Spanish Ministry of Economy and Competitiveness (MINECO) via competitive grants CGL2013-46808-R and CGL2017-89149-C2-1-R. AH acknowledges funding from EPSRC Fellowship EP/N030141/1.

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

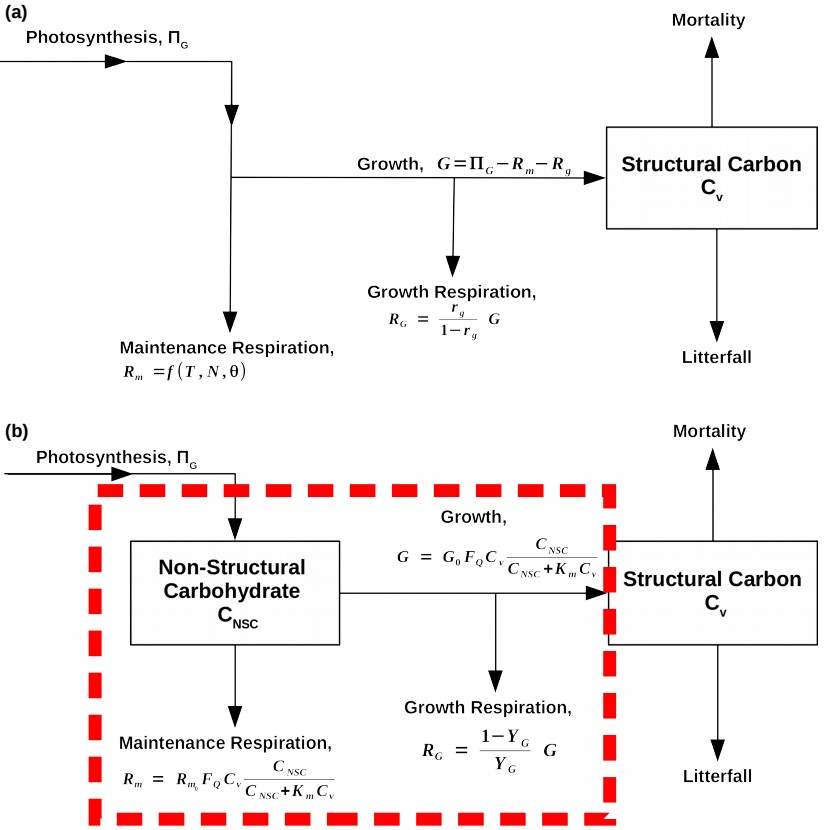

**Figure 1.** Flow diagrams that demonstrate how SUGAR is designed to change the model structure of carbon allocation within the Joint UK Land Environment Simulator (JULES) (Best et al., 2011; Clark et al., 2011)). Arrows represent fluxes of carbon and black boxes represent carbon pools. **(a)** A representation of the current structure of carbon allocation in JULES. Maintenance respiration ($R_m$) depends on temperature (T), leaf nitrogen (N) and optionally, water availability ($\theta$). Growth respiration ($R_G$) is equal to a constant fraction of growth ($G$) which is equal to photosynthesis ($\Pi_G$) less total plant respiration ($R_G + R_m$). Total utilisation of carbon ($R_m + R_G + G$) is always exactly equal to carbon assimilation by photosynthesis ($\Pi_G$). **(b)** A representation of how SUGAR would sit within JULES. The red dashed box represents the model boundary of SUGAR. Both maintenance respiration and growth depend on temperature via a $Q_{10}$ function ($F_Q$), structural biomass ($C_v$) and non-structural carbohydrate content ($C_{NSC}$). Growth respiration is a constant fraction of growth.

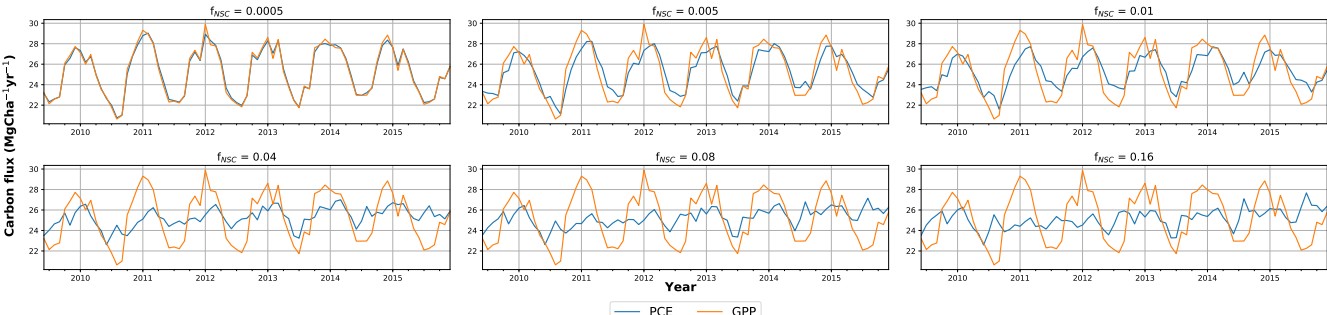

**Figure 2.** Simulated plant carbon expenditure (PCE) from SUGAR against gross primary productivity (GPP) (Parazoo et al., 2014) for different initialised carbohydrate content as a fraction ($f_{NSC}$) of grid-box biomass (Avitabile et al., 2016).

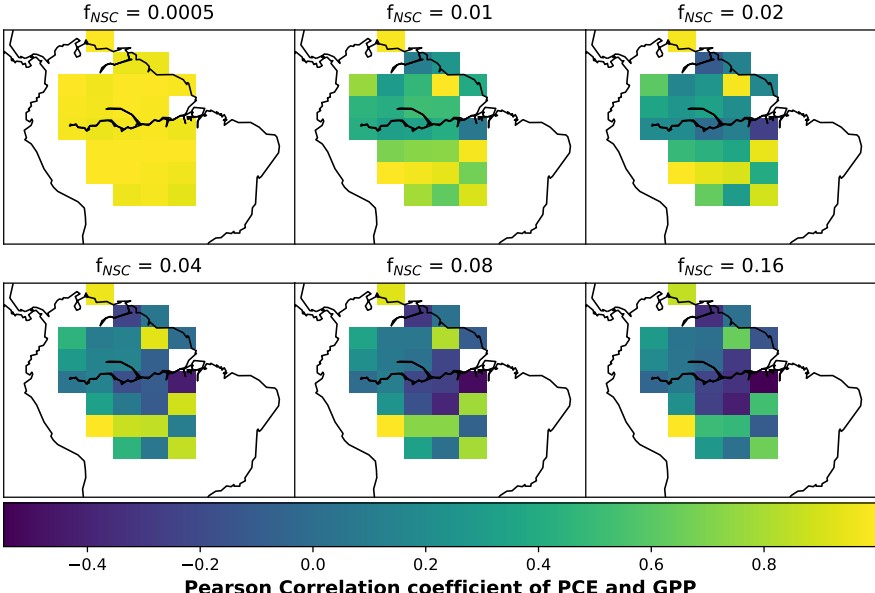

**Figure 3.** The Pearson correlation coefficient of simulated plant carbon expenditure (PCE) and driving gross primary productivity (GPP) for different initialised carbohydrate contents as a fraction ($f_{NSC}$) of grid-box biomass. This gives an indication of how important a driver GPP is for PCE in each grid-box.

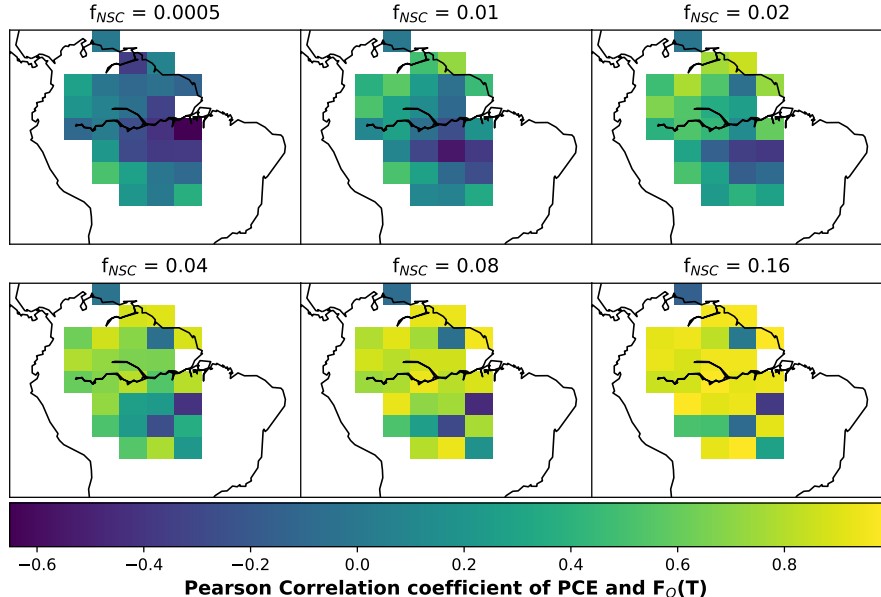

**Figure 4.** The Pearson correlation coefficient of simulated plant carbon expenditure (PCE) and driving $Q_{10}$ ($F_Q$) for different initialised carbohydrate contents as a fraction ($f_{NSC}$) of grid-box biomass. This gives an indication of how important a driver the $Q_{10}$ function is for PCE in each grid-box.

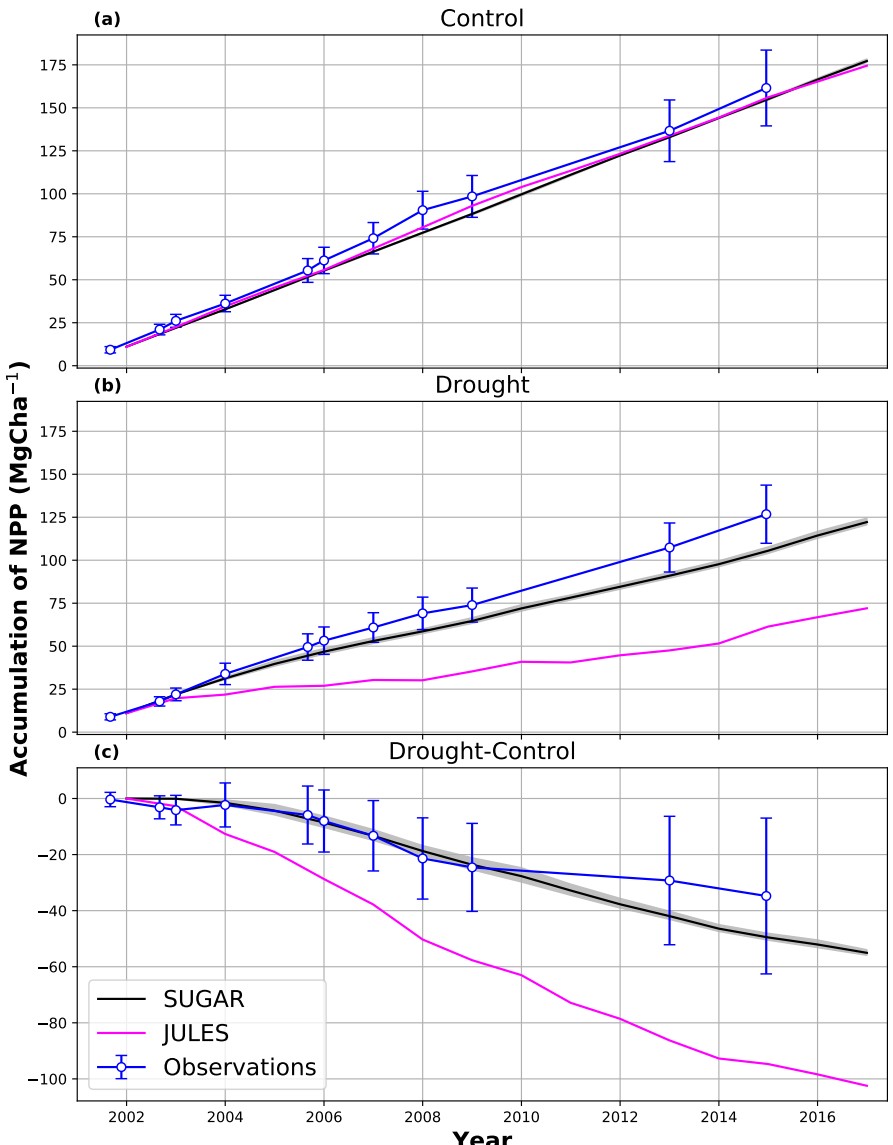

**Figure 5.** Accumulated Net Primary Productivity at Caxiuanã in (a) Control plot, (b) TFE plot and (c) The difference between the drought and control forest (TFE-control). Observations are calculated as the accumulated sum of above-ground biomass increment change and total local litter-fall (Rowland et al., 2018). The presented confidence intervals are the sum of the litterfall measurement error and the 95% confidence intervals of biomass increment calculated from 8 allometric equations using trunk diameter at breast height (DBH) data from Caxiuanã. The uncertainty envelope on SUGAR represents the maximum and minimum of an ensemble of simulations in which parameter $a_{K_m}$ was varied between 0.1 and 2.0.

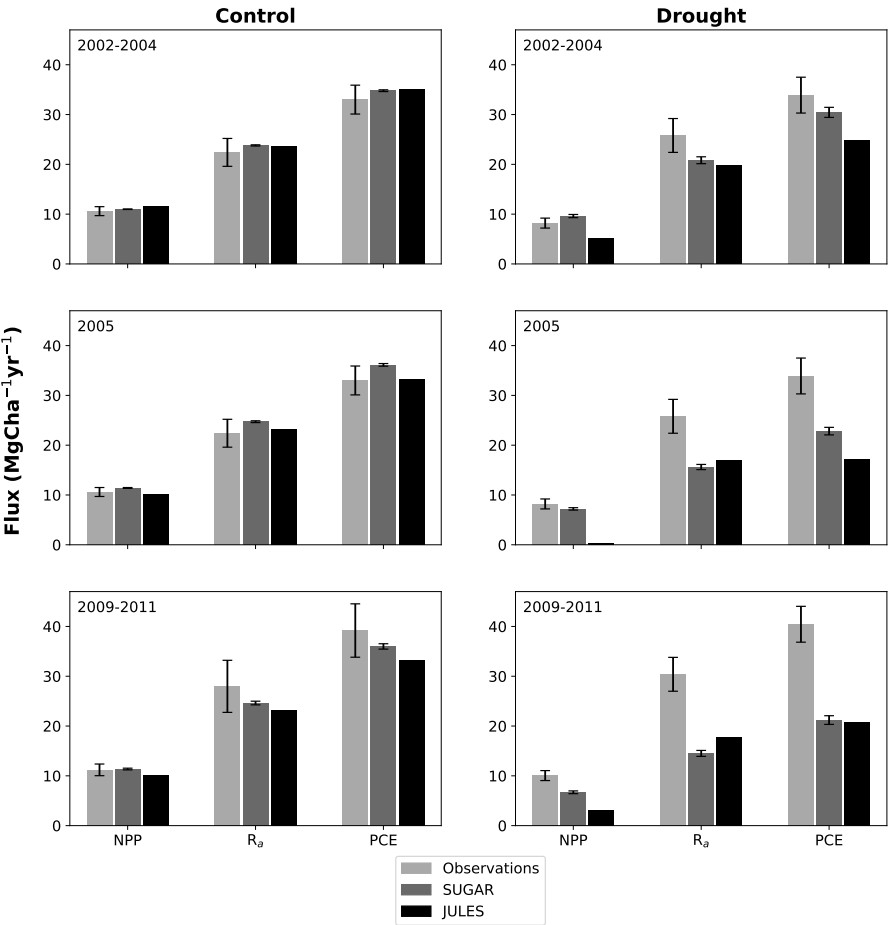

**Figure 6.** Net primary productivity (NPP), Autotrophic respiration ($R_p$) and Plant Carbon Expenditure (PCE = NPP+$R_p$); for the periods 2002-2004, 2005 and 2009-2011. The left column is from the control plot and the right is from the through-fall exclusion (TFE) plot. Model predictions from JULES and SUGAR are calculated by taking the mean of each flux over each period. Observations for 2005 are from Metcalfe et al. (2010) and observations from 2009-2011 are from da Costa et al. (2014). Simulated photosynthesis in JULES responded almost instantly to the introduction of the panels on the TFE plot which meant that NPP, $R_p$ and PCE changed significantly in both models between 2002 and 2005. To demonstrate this change we show predicted fluxes during the 2002-2004 period as well as from 2005. Observations for this period are not available to such a comprehensive degree as they are for 2005 and the 2009-2011 period. For this reason we compare the model predictions for 2002-2004 to the 2005 observations. This is reasonable in the control plot where it is plausible that the forest was in steady state (Metcalfe et al., 2010) and so fluxes from 2005 will be similar to those during the 2002-2004 period. In the TFE plot while there were some significant changes in observed carbon fluxes during the first 3 years of the experiment, (for example the production of leaves, flowers and fruits, and fine wood (Rowland et al., 2018; Meir et al., 2018)), the forest largely resisted the effects of the drought during this period (significant increases in mortality were not seen until 2005 (Rowland et al., 2015; Meir et al., 2018)) and so we can similarly expect fluxes from 2002-2004 to be comparable to those from 2005. Nonetheless, care should be taken with these comparisons in both plots. The error bars on SUGAR represent the maximum and minimum of an ensemble of simulations in which parameter $a_{K_m}$ was varied between 0.1 and 2.0.

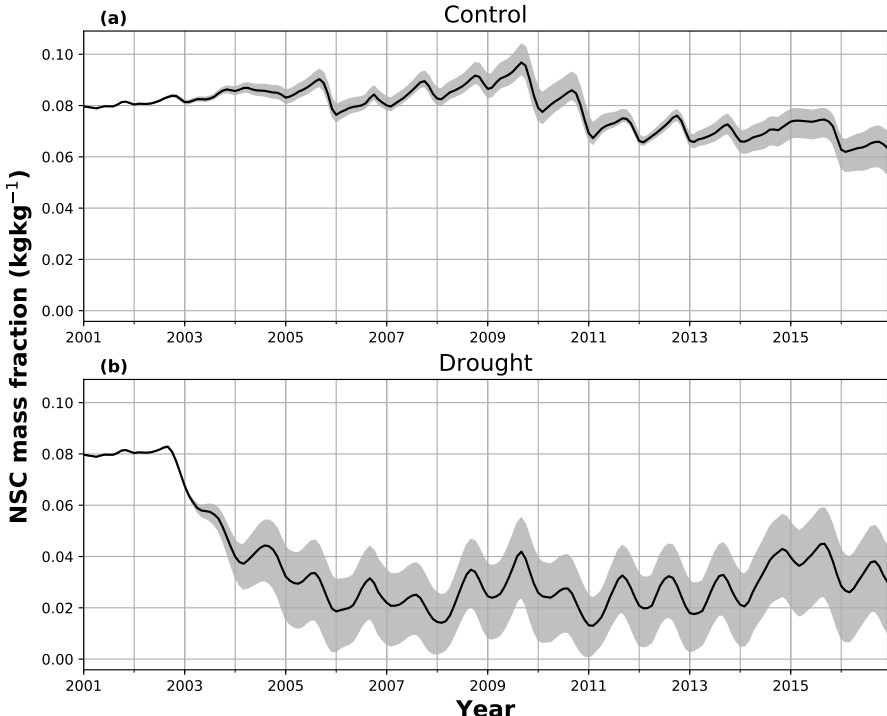

**Figure 7.** The effect of the parameter $a_{K_m}$ in SUGAR on simulated non-structural carbohydrate (NSC) as a fraction of total carbon biomass, in (a) the control plot and (b) the TFE plot. The mean, maximum and minimum from an ensemble of simulations where $a_{K_m}$ is varied between 0.1 and 2.0 are presented.

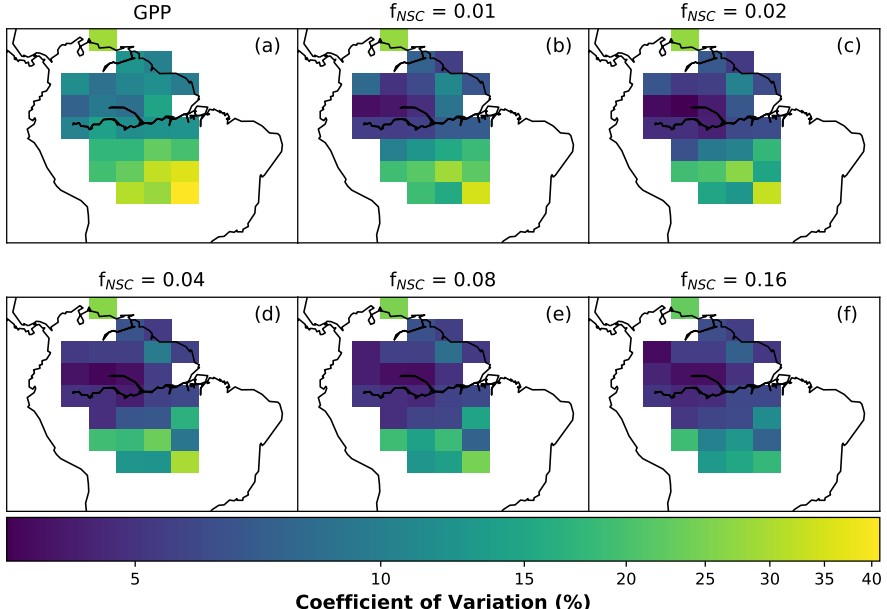

**Figure A1.** The coefficient of variation of **(a)** GPP (Parazoo et al., 2014) and **(b-f)** simulated Plant Carbon Expenditure (PCE) for different initialised carbohydrate content as a fraction of grid-box Biomass ($f_{NSC}$).

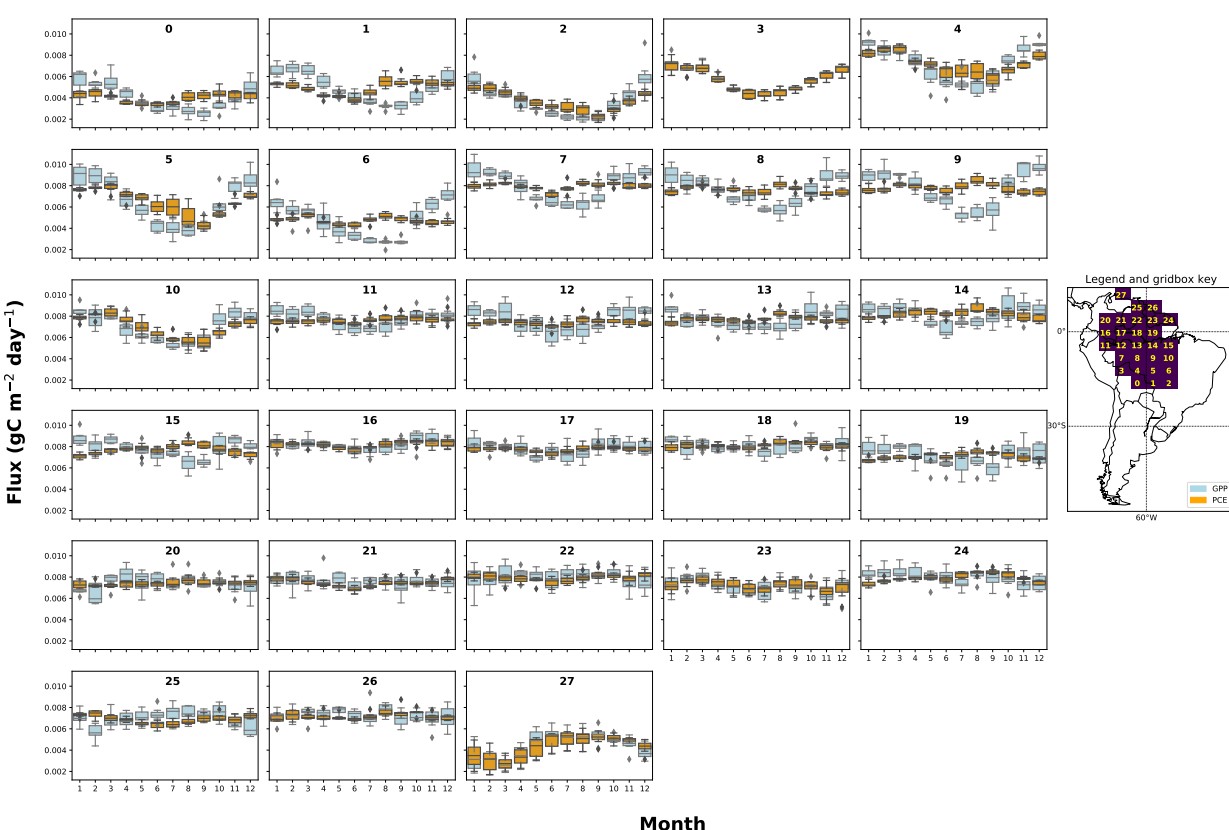

**Figure A2.** The mean seasonal trend of simulated plant carbon expenditure (PCE) and forcing gross primary productivity (GPP) (Parazoo et al., 2014) for each gridbox in the $f_{NSC}$ =0.08 SUGAR simulations. The map key shows which plot corresponds to which grid-box.

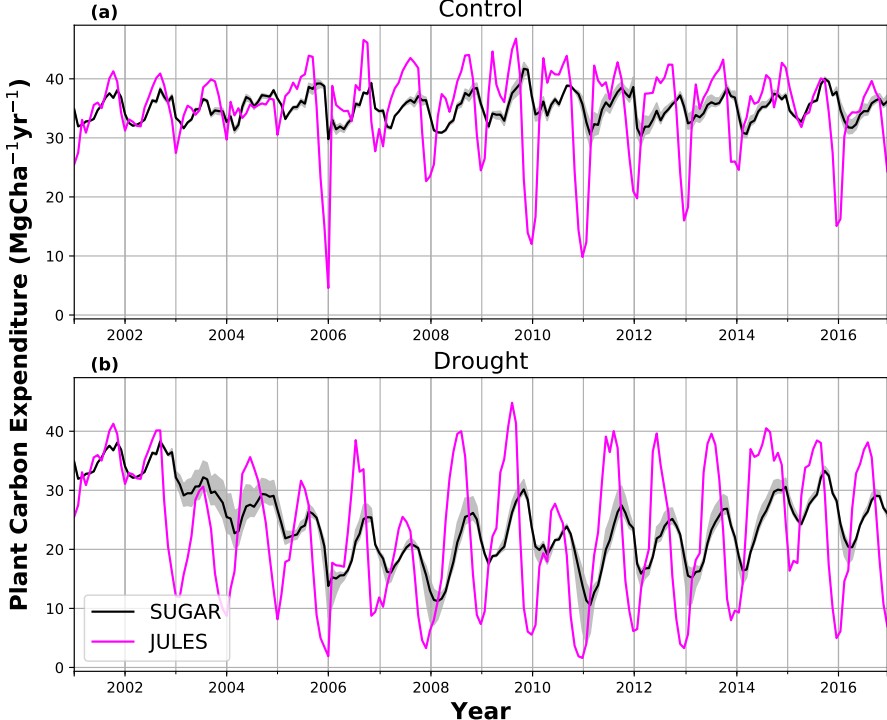

**Figure A3.** Simulated plant carbon expenditure (PCE) from JULES and SUGAR for (a) the control and (b) the through-fall exclusion (TFE) plots at Caxiuanã. A sensitivity study on the parameter $a_{K_m}$ in SUGAR was carried out and the maximum, minimum and ensemble mean PCE are presented. Time-series observations of PCE from the site were not available.

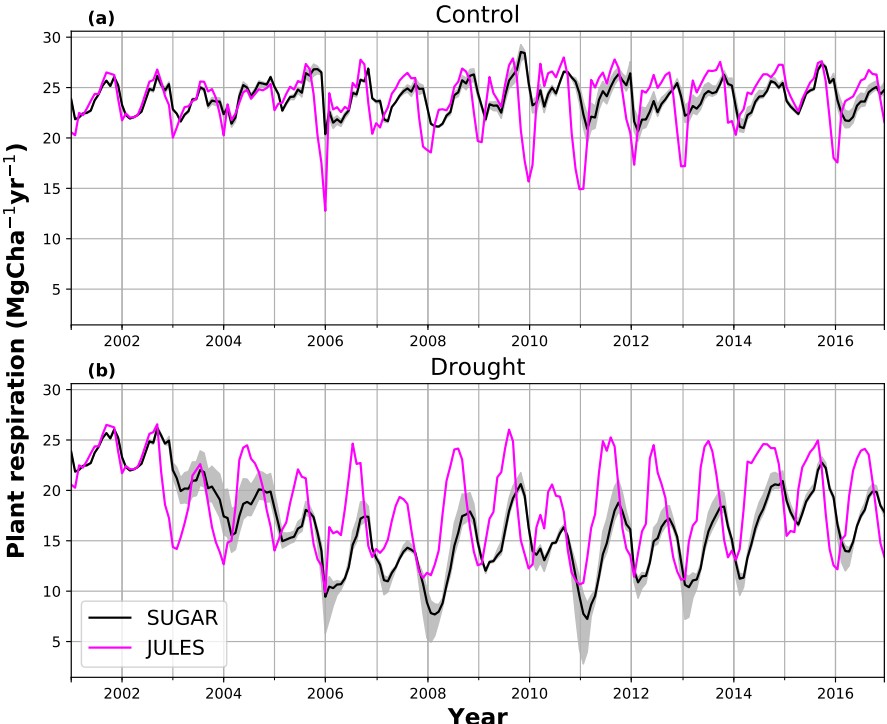

**Figure A4.** Simulated plant respiration (R) from JULES and SUGAR for (a) the control and (b) the through-fall exclusion (TFE) plots at Caxiuanã. A sensitivity study on the parameter $a_{K_m}$ in SUGAR was carried out and the maximum, minimum and ensemble mean R are presented. Time-series observations of R from the site were not available.

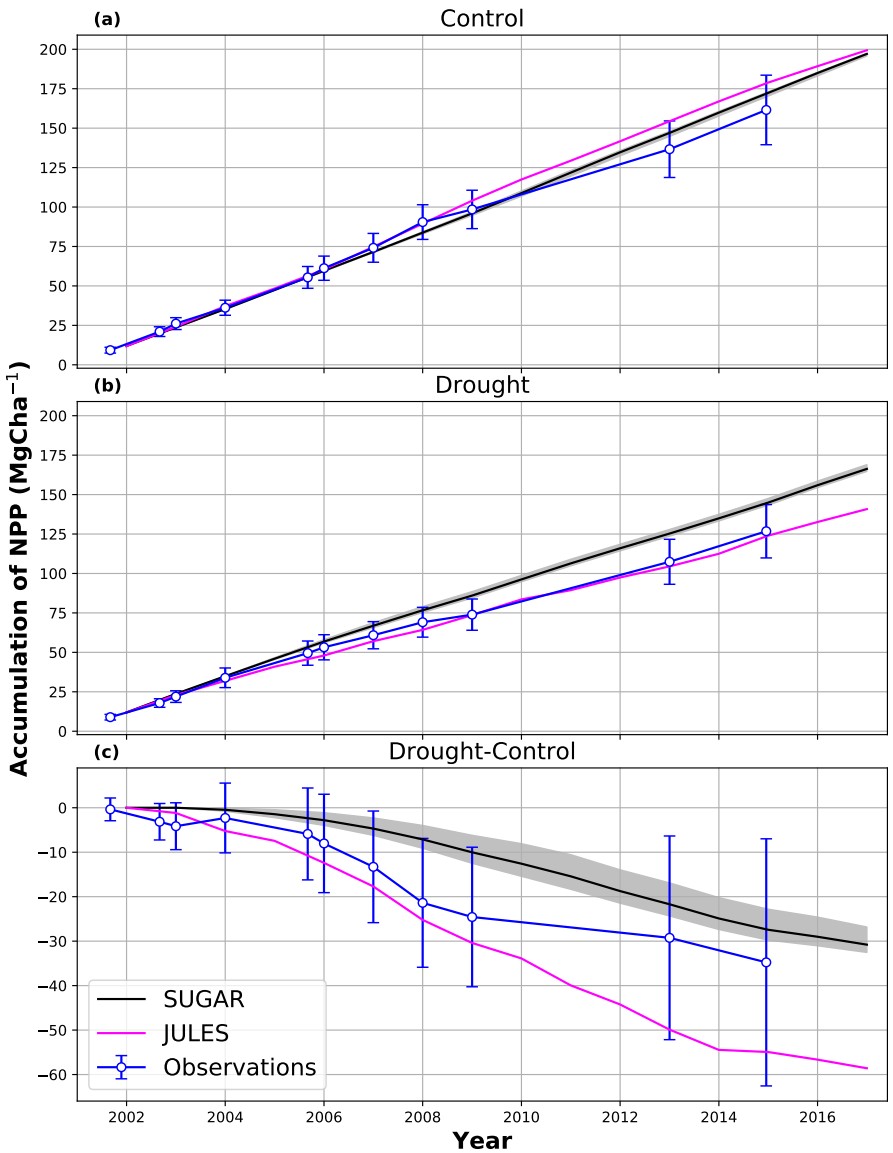

**Figure A5.** Accumulated Net Primary Productivity at Caxiuanã in (a) Control plot, (b) TFE plot and (c) The difference between the drought and control forest (TFE-control). Soil moisture stress has been artificially reduced in JULES by 50% and the resulting GPP has been used to drive SUGAR. Observations are calculated as the accumulated sum of biomass increment change and local litter-fall (Rowland et al., 2018). The presented confidence intervals are the sum of the litterfall measurement error and the 95% confidence intervals of biomass increment calculated from 8 allometric equations using trunk diameter at breast height (DBH) data from Caxiuanã.

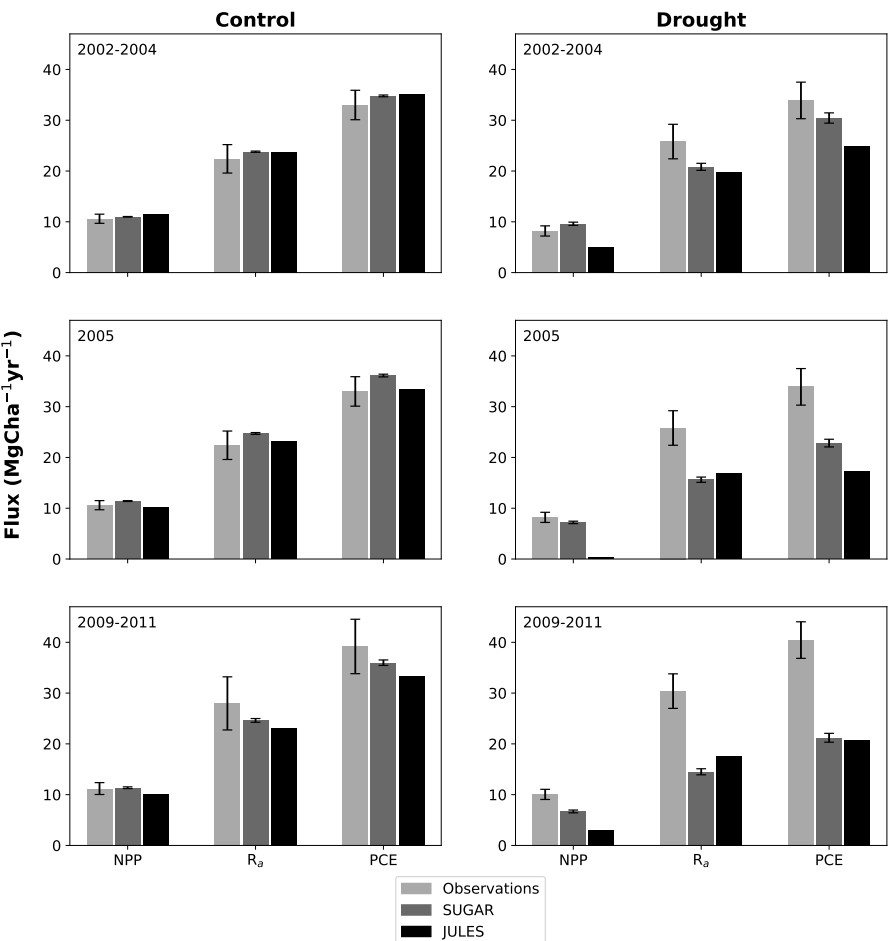

**Figure A6.** Net primary productivity (NPP), Autotrophic respiration ($R_p$) and Plant Carbon Expenditure (PCE = NPP+$R_p$); for the periods 2002-2004, 2005 and 2009-2011. The left column is from the control plot and the right is from the through-fall exclusion (TFE) plot. Soil moisture stress has been artificially reduced in JULES by 50% and the resulting GPP has been used to drive SUGAR. Model predictions from JULES and SUGAR are calculated by taking the mean of each flux over each period. Observations for 2005 are from Metcalfe et al. (2010) and observations from 2009-2011 are from da Costa et al. (2014). Simulated photosynthesis in JULES responded almost instantly to the introduction of the panels on the TFE plot which meant that NPP, $R_p$ and PCE changed significantly in both models between 2002 and 2005. To demonstrate this change we show predicted fluxes during the 2002-2004 period as well as from 2005. Observations for this period are not available to such a comprehensive degree as they are for 2005 and the 2009-2011 period. For this reason we compare the model predictions for 2002-2004 to the 2005 observations. This is reasonable in the control plot where it is plausible that the forest was in steady state (Metcalfe et al., 2010) and so fluxes from 2005 will be similar to those during the 2002-2004 period. In the TFE plot while there were some significant changes in observed carbon fluxes during the first 3 years of the experiment, (for example the production of leaves, flowers and fruits, and fine wood (Rowland et al., 2018; Meir et al., 2018)), the forest largely resisted the effects of the drought during this period (significant increases in mortality were not seen until 2005 (Rowland et al., 2015; Meir et al., 2018)) and so we can similarly expect fluxes from 2002-2004 to be comparable to those from 2005. Nonetheless, care should be taken with these comparisons in both plots.

| Symbol | Units | Definition |
|---|---|---|
| $a_{K_m}$ | | Saturation parameter |
| $C_{NSC}$ | kg C m$^{-2}$ | NSC content |
| $C_v$ | kg C m$^{-2}$ | Structural carbon content |
| $f_{NSC}$ | | Equilibrium NSC mass fraction |
| $F_Q$ | | $Q_{10}$ function for growth and respiration |
| $G$ | kg C m$^{-2}$ s$^{-1}$ | Plant growth |
| $G_0$ | s$^{-1}$ | Specific growth rate |
| $q_{10}$ | | $Q_{10}$ value for plant respiration and growth |
| $R_g$ | kg C m$^{-2}$ s$^{-1}$ | Growth respiration |
| $R_m$ | kg C m$^{-2}$ s$^{-1}$ | Maintenance respiration |
| $R_{m_0}$ | s$^{-1}$ | Specific rate of maintenance respiration |
| $R_p$ | kg C m$^{-2}$ s$^{-1}$ | Total plant respiration |
| $T$ | $^\circ$C | Temperature |
| $U$ | kg C m$^{-2}$ s$^{-1}$ | Plant carbon expenditure |
| $Y_g$ | | Growth yield coefficient |
| $\alpha$ | | Ratio of plant growth to PCE |
| $\Pi$ | kg C m$^{-2}$ s$^{-1}$ | Net primary productivity |
| $\Pi_G$ | kg C m$^{-2}$ s$^{-1}$ | Gross primary productivity |
| $\tau$ | s | Ecosystem carbon residency time |
| $\phi$ | s$^{-1}$ | Specific rate of carbohydrate utilisation |

**Table 1.** Definitions of Symbols

| Parameter | Units | Value (Cax) | Range | Description | Justification |
|---|---|---|---|---|---|
| $f_{NSC}$ | kg kg$^{-1}$ | 0.16 | 0.1-0.4 | Equilibrium NSC mass fraction | (Wurth et al 2005) |
| $q_{10}$ | | 2.0 | | Factor by which respiration and growth increase given a 10 degree warming | (Ryan 1991) |
| $Y_g$ | | 0.75 | | Growth conversion efficiency | (Thornley and Johnson 1990) |
| $a_{K_m}$ | | 0.1-2.0 | 0.1-2.0 | Relates the half saturation NSC mass fraction ($K_m$) with the equilibrium pool size ($f_{NSC}$) | Sensitivity study carried out in this study |
| $\alpha$ | | 0.32 | 0.3-0.5 | Ratio of plant growth to total carbohydrate utilisation | Evaluated by setting equal to steady state carbon use efficiency (CUE*). Between 0.3-0.5 for a tropical forest (Chambers et al., 2004; Gifford, 1995) |
| $\phi$ | yr | $5.15(1 + a_{K_m})$ | | Maximum specific rate of NSC utilisation at 25°C | Evaluated in terms of $a_{K_m}$ using average specific photosynthesis $\left(\frac{\Pi_G}{C_v}\right)^*$ and temperature ($T$) of a forest in steady state. $\phi = \frac{1 + a_{K_m}}{F_Q^*(T)}\left(\frac{\Pi_G}{C_v}\right)^*$. Can also be evaluated in terms of vegetation carbon residency time $\tau$ (e.g. Carvalhais et al., 2014): $\phi = \frac{1 + a_{K_m}}{F_Q^*(T)\tau}$ |

**Table 2.** Parameters in SUGAR

$f_{NSC}$ the is fraction of NSC relative to total structural carbon and so estimates of NSC as a fraction of total dry mass should be adjusted to account for non-carbon biomass.

| Author | Equation | a | b | c | d | E |
|---|---|---|---|---|---|---|
| Brown (1997)a | $a + bD + cD^2$ | 42.69 | -12.8 | 1.242 | | |
| Brown (1997)b | $\exp(a + b\log_e(D))$ | -2.134 | 2.53 | | | |
| Carvalho Jr. et al. (1998) | $1000a\exp(b + c\log_e(D/100))$ | 0.6 | 3.323 | 2.546 | | |
| Araújo et al. (1999) | $abD^c$ | 0.6 | 4.06 | 1.76 | | |
| Chambers et al. (2001) | $\exp(a + b\log_e(D) + c\log_e(D)^2 + d\log(D)^3)$ | -0.37 | 0.333 | 0.933 | -0.122 | |
| Baker et al. (2004) | $\exp(a + b\log_e(D) + c\log_e(D)^2 + d\log(D)^3)(\rho/0.67)$ | -0.37 | 0.333 | 0.933 | -0.122 | |
| Chave et al. (2005) | $\exp(a + b\log_e(D) + c\log_e(D)^2 + d\log(D)^3)(\rho)$ | -1.499 | 2.148 | 0.207 | -0.0281 | |
| Chave et al. (2014) | $\exp(a - 0.976E + b\log_e(D) + c\log_e(D)^2 + d\log(\rho))$ | -1.803 | 2.673 | -0.0299 | 0.976 | -0.0510307 |

D = Diameter at breast height (dbh); $\rho$ = Wood density; a, b, c, d, E are constants.

**Table 3.** Allometric equations used to calculate above-ground biomass, $C_v$ (kg)