# Peer review of "The Impact of a Simple Representation of Non-Structural Carbohydrates on the Simulated Response of Tropical Forests to Drought"

_Biogeosciences, 2019_

## Referee Comment (RC1) · Martin De Kauwe (Referee) · 19 Dec 2019

Jones et al. present a new storage model that could be readily incorporated into LSMs, allowing them to decouple growth from photosynthesis in line with emerging experimental evidence. This manuscript tackles a very interesting scientific question, is well written and the results are clearly presented. I think with revision this manuscript will make a nice contribution to the literature. Below I've outlined a number of small issues. I have a few larger issues with the current manuscript.

1. I felt the details on how the model was parameterised could be improved. I strongly felt the paper needs a table with parameters used, which would make this paper repeatable.

2. I felt a number of the plots were a little redundant. In fact, if you run their code, I'd say the timeseries of NSC, perhaps with the addition of changing water availability, would be more insightful as to how the model works.

3. I was a little bothered about how different the implementation of NSC actually was from LSMs that assume excess GPP goes into a labile pool, which is then used for growth/respiration? If I'm doing the authors a disservice here then I apologise, but perhaps a few more words outlining this distinction are required. I guess the bigger point I'm making here is that I was anticipating clear hypotheses about *how* and *when* such a labile pool would be used. I do not see these. For example, does the plant aim to maintain a minimum labile pool? What sets this? How big a pool does it accumulate? How would these things vary between PFTs? In the same way, what about the timing of utilisation? The authors spend the introduction sets up a clear link to water stress and this model as a plausible buffering mechanism. But the treatment of water stress in the manuscript is insufficient. It occurs to me that the authors are assuming that a plant will regulate GPP in the same way with and without a NSC pool (this is implied by the offline implementation). But does this make sense? The details are not given, but presumably in JULES water stress reduced the assimilation rate via reducing Vcmax. But if you have a NSC pool, would that imply a plant might be a little riskier? If it has some stockpile, why be quite so sensitive to water availability? I have nothing to support this line of thinking, but it seems pretty testable and logical (if only to me!).

4. As I said above, the results are pretty convincing, but they are also indirect. There is nothing to support the model results being for the right mechanistic reasons. We have nothing that shows us the NSC timeseries (not shown) is supported experimentally from the throughfall experiment. We have nothing to say the respiration from this pool is supported experimentally. In both cases, I suspect a reader will anticipate such plots, I certainly did. Do such data exist? I have no idea if they do or not.

5. Finally, the authors put forward an argument that the NSC model results at the throughfall experiment are limited by the poor representation of water stress on GPP in JULES. This is testable. All the authors have to do is make the GPP reduction less sensitive to water stress and plug these GPP values into their model. My suspicion is that the agreement with the obs may not improve, but I might be wrong. It would be worth testing this rather than speculating.

Merry Christmas,

Martin De Kauwe

Introduction ————

* To be honest, I found the whole first paragraph completely unrelated to the focus of the paper. I think the paper would really benefit from a more relevant opening paragraph, entirely up to the authors what they do here, just a suggestion.

* Pg 2, ln 24: "and so plants rely heavily on their NSC reserves". Are there any numbers to support this statement? How heavily? For how long?

* Pg 3, ln 41: what is the evidence for carbon starvation leading to mortality? My understanding is that it is essentially non-existent outside of a few potted experiments? See for example, Adams et al. A multi-species synthesis of physiological mechanisms in drought-induced tree mortality. I think this could be more carefully phrased.

* Pg 3, Second paragraph "Despite...". Obviously biased, and feel free to ignore...but I will draw your attention to "Mahmud et al. Inferring the effects of sink strength on plant carbon balance processes from experimental measurements. Biogeosciences.", which I think is a nice attempt to exploit experimental data to help mechanistically unpick the role of sink control. I highlight this paper because the focus was specifically to aid model development - "This can largely be attributed to a scarcity of ecosystem-level data (NSC content and distribution) that can be used to parametrise and evaluate models for a range of species and climates that covers all plant functional types (PFTs)

used in LSMs".

Methods ———-

* Is there any supporting evidence for the assumptions in SUGAR? It's is fine if there isn't but it might be nice to cite some relevant literature if there is. For example, section 2.2 ...

* In 2.5, it would be useful for the parameter ranges to be given to the reader? The section is titled parameter estimation but I've got no idea after reading it what values were used. I think a table with assumed parameters and/or ranges would be very useful for a reader who wished to repeat any of this. Currently, the only defined terms are the Q10 and Yg. For example, in the results: "All other parameters (Yg, aKm , q10) are kept constant at their default values (see model description)." Where was the value of aKm given?

* I think the methods would benefit from a few sentences/paragraph explaining how the model works beyond the equations. Most of the introduction set up an interpretation of the use of NSC during periods of water stress but this theme is not returned to once in the methods. How does water stress interact here? It clearly isn't directly, but just comes about due to growth demand? What about the timescales of utilisation or storage increase? My reading of the model is that there is no specific hypothesis being tested here about increases in NSC. It is simply the difference between C uptake and utilisation. I don't really see that this goes beyond what many LSMs currently assume with respect to a excess carbon storage pool. I was expected a hypothesis about how plants might aim to maintain a storage pool, which I do not see. Equally, something about how they might prioritise a draw-down of this pool.

* Pg 6: "optimised so that annual GPP and NPP in the control forest agree with observations." What specifically was optimised here?

* Surely the SUGAR model was simple enough to just embed in JULES. Some further

explanation is warranted here as to why this wasn't just done...

Results ——-

* As a general statement, I found it odd to start with the spatial interpretation rather than a site-level analysis. Doing it this way round is harder to see how the model is really working and to me (at least), it would make more sense to reverse the presentation of the results. Or alternatively, it would be useful to see a timeseries extracted from Fig 2. For example, "This decline in seasonal variation is caused by an increase in dry season carbon expenditure and a decrease in the wet season carbon expenditure.". It would be nice to see this...

* I understand Fig 2 is a sensitivity experiment, but how are we meant to interpret whether the SUGAR model is improving / degrading growth predictions? I can see that increasing the fSNC dampens the variation, but is this dampening supported in any way? I wonder if Fig 3 is strictly necessary? It seems to be implied by Fig 2, I feel like you need one or the other. Perhaps it is a supplementary figure. I'd much rather a few timeseries plots!

* The implication of Fig 4 is that the respiration assumption in the model becomes more important as fNSC increases. How sensible is the assumed respiration eqn...this seems quite important.

* In the first paragraph of 4.2, it would be good to explain why JULES NPP and SUGAR NPP differ at all? If there is no water stress where does this difference come from? Do the models have different respiration assumptions? A shift in timescale of growth?

* Following the sensitivity experiment in Fig 2 onwards, my interpretation was that it was therefore not obvious how to parameterise the model. As such, I was expecting to see some form of uncertainty envelope around the SUGAR model line in Fig 5? How was SUGAR parameterised in this set of runs? I found this very unclear in my head at this point of the manuscript.

* With Fig 5, arguably you don't need panel b, you could perhaps then include a respiration comparison between JULES and SUGAR? No idea what that looks like...

---

## Referee Comment (RC2) · Thomas Pugh (Referee) · 5 Feb 2020

Jones et al. describe a simple model of NSC dynamics which is aimed at incorporation in large-scale ecosystem models, particularly JULES. They evaluate the model based on observations at a throughfall exclusion experiment in the Amazon and provide a demonstration of how the new model formulation affects carbon fluxes at the scale of the Amazon. Dealing with NSC storage dynamics in large-scale models is of clear conceptual importance for capturing tree mortality and intra-annual carbon fluxes, but presents a significant challenge given the lack of parameterisation data. Simple, flexible representations of NSC dynamics, such as that presented here, are therefore

needed to underlie progress in ecosystem model development as knowledge of this topic develops. The paper is very well written, and the introduction in particular is excellent, giving a clear, well-structured and thorough overview of the topic and problem. Overall, this seems a very nice development, but I was particularly missing discussion/testing of some important assumptions.

Reflecting the lack of observations of NSC stocks and dynamics, the model is designed to be parameterised based on more commonly-available measurements such as biomass and GPP. This is an elegant idea, however some fairly substantial assumptions are made to achieve it, and those assumptions are not well explored here. One key assumption, made in Eq. 8, is that the NSC dependence of maintenance respiration is the same as that for plant growth. The implicit hypothesis is that respiration and growth are equally prioritised under resource limitation. As far as I am aware, there is no clear evidence to support this and it is just as likely that the plant should down-regulate growth to maintain respiration-related functions. No reasoning or citations are presented to support this key choice. In my opinion a thorough discussion is necessary.

I am also a bit curious about the assumption implied in Eq. 3 that NSC in the heartwood is available to trees to use. Is it posited that there are mechanisms that trees can use to extract this NSC from dead wood, in which case citations are needed, or is this simply an assumption to enable use of the widely available total biomass information? The latter would seem perfectly reasonable to me, but then does seem to warrant a short discussion about the fraction of total NSC that is typically found in heartwood (e.g. <15%, Richardson et al., 2015).

Similarly, there has been clear effort to minimise the number of parameters, but choices of parameter values are not well justified in the text. Parameters $a_{Km}$ and $q_{10}$ are assigned default values with no justification given for the choice. In particular, $a_{Km}$ appears to be central to the results. If clear, strong, justification for the choices cannot be given, then the sensitivity of the results to these parameters should be included in the tests. The parameter $f_{NSC}$ is appropriately treated as a range, although I would

have been inclined to set upper limit of the range a bit above that actually observed in a tropical forest to explore the parameter space a bit, but fair enough!

Pg. 4, lines 11-12. So this means that SUGAR assumes that trees actively allocate to storage to maintain a certain store size. Perhaps acknowledge this decision explicitly here, given the active debate on this (which was introduced in the introduction)?

Pg. 7, line 8. Is this above-ground NPP or total NPP? If the former, how is the output from SUGAR being adjusted to compensate for this in the evaluation?

Pg. 7, line 16. How is the CV calculated? Annual mean of daily (or monthly) values, followed by taking the mean across the simulated years? Or directly over the whole dataset? I'm trying to understand if this is showing intra- or interannual variability.

Pg. 8, line 31. Doesn't this imply that the NSC store has been underestimated? Perhaps worth exploring how much storage would be required to maintain respiration and growth and how this is affected by parameters like $a_{Km}$? This could provide a useful hypothesis for future investigation.

Pg. 8, lines 42-44. Maybe a bit of overinterpretation of small differences here? Overall JULES and SUGAR seem equally good for the control plots in Figs. 5 and 6.

Pg. 8, line 51. Whilst the downregulation of respiration in response to depleting NSC may help buffer NPP, it clearly doesn't improve the simulation of respiration, which is more strongly underestimated in SUGAR than in JULES from 2005 onwards. I think this discussion needs to reflect that whilst SUGAR provides a very useful representation of a process we are confident is important, including this process does not by itself radically improve the overall carbon flux simulation for the drought experiment investigated.

Fig. 5. It would be helpful to also see timeseries of respiration and NSC storage to allow a full and balanced interpretation.

References

Richardson et al. (2015) Distribution and mixing of old and new nonstructural carbon in two temperate trees, New Phytologist 206(2), 590-597.

---

## Author Comment (AC1) · 26 Mar 2020

[bg]copernicus xcolor

1. I felt the details on how the model was parametrised could be improved. I strongly felt the paper needs a table with parameters used, which would make this paper repeatable.

This is a good suggestion, and we agree that the description of model parametrisation, in general, is not sufficiently clear. We have added the following table to show all the parameters in SUGAR, their ranges, and values used in the Caxiuanã simulations. We have also updated the derivation of the parameter evaluation process which we also include below. This includes details on how to parametrise SUGAR for single site runs, where initialisation data from the site is available (i.e. as in the Caxiuanã simulations) and how SUGAR might be parametrised for more general or global simulations. We have also outlined the distinction between running SUGAR offline and coupling it to a land surface model (LSM), which is not really clear in the current manuscript. Hopefully this will make it clearer how we have evaluated the parameters in these simulations and how to apply SUGAR to other experiments.

To address the point regarding parameter evaluation we outline below the complete process, including a discussion of what data can be used depending on the nature of the simulation being conducted. We will happily add this section to the main body of the paper between the model description and methods section. We will also update the methods section to more clearly outline how this process is applied to both experiments in the paper. Note, for the purposes of this response we include below the definitions of symbols, however, in the updated manuscript we would omit these since they are defined in the model description section.

**Parameter evaluation - derivation**

First we address parameters that have standard or commonly used values within the land surface modelling literature, and parameters that can be directly estimated from empirical data. These are $q_{10}$ and $Y_g$, and $f_{NSC}$ respectively.

$q_{10}$

The Q10 function is commonly used to describe the temperature dependence of plant respiration and growth in LSMs. Growth and respiration both depend on temperature via a Q10 function with the standard $q_{10}$ parameter value of 2.0 (Ryan 1991).

$Y_g$

The $Y_g$ parameter is derived in Thornley and Johnson (1990), and estimated to equal 0.75. $Y_g$ or similar parameters are used in LSMs (e.g Clark et al. 2011), often with an equivalent value of 0.75.

$f_{NSC}$

The $f_{NSC}$ parameter represents the equilibrium NSC pool size as a fraction of total structural carbon. This can be set directly using empirical data e.g. for tropical forests using Wurth et al. (2005). Note that studies such as Wurth et al. (2005) present NSC stocks as a fraction of total dry mass and so data should be adjusted to account for non-carbon mass.

$a_{K_m}$
It is not possible to directly evaluate $a_{K_m}$ from empirical data. It is given the default value of 0.5, since this gives numerically stable results on the commonly used time-step of 1 hour, and the dependence of respiration and growth on NSC begins to saturate at reasonable NSC mass fractions.

The remaining parameters, $\phi$ and $\alpha$, cannot be evaluated directly using empirical data so below we outline how the model equations can be manipulated to find $\phi$

and $\alpha$ in terms of quantities that can be estimated using observations.

We start by finding the rate of change of NSC mass fraction, $W_{NSC}$, where $W_{NSC} = C_{NSC}C_v$, in terms of $C_{NSC}$ and $C_v$:

$$dW_{NSC}dt = 1C_v dC_{NSC}dt - W_{NSC}1C_v dC_v dt \qquad (1)$$

We then consider the case where the NSC mass fraction is constant and the left hand side of equation (1) is zero. In reality the NSC mass fraction of forest will not be exactly constant and variations in environmental variables will cause changes in NSC stocks. However, for a non-stressed forest it is a good assumption that over a prolonged period, $\tau_{stable}$, the NSC mass fraction will be roughly constant. For example, we can assume that over the course of one year, a non-stressed forest will use as much carbon as it assimilates and consequently will end the year with roughly the same NSC stock with which it started. This means that we can integrate equation (1) over this period and set the left hand side equal to zero:

$$0 = \int_{\tau_{stable}} \left(1C_v dC_{NSC}dt - W_{NSC}1C_v dC_v dt\right) dt \qquad (2)$$

We then consider two cases:

1. **Uncoupled simulations**
   When SUGAR is run offline, i.e. it is not coupled to a LSM, the $dC_v dt$ term must be neglected and we assume $C_v$ is constant. Hence:

$$dW_{NSC}dt = 1C_v dC_{NSC}dt \qquad (3)$$

We then use the equation (1 in the original manuscript) for the rate of change of NSC:

$$0 = \int_{\tau_{stable}} \left(\Pi_G C_v - RC_v - GC_v\right) dt \qquad (4)$$

To evaluate $\phi$ we use the equation for total carbohydrate utilisation and rearrange:

$$\phi \int_{\tau_{stable}} F_Q(T) W_{NSC} W_{NSC} + K_m dt = \int_{\tau_{stable}} \Pi_G C_v dt \qquad (5)$$

We divide both sides by $\tau_{stable}$ and assume that this can be approximated as:

$$\phi F_Q^*(T) W_{NSC}^* W_{NSC}^* + K_m = (\Pi_G C_v)^* \qquad (6)$$

Where the asterisk denotes a temporal average over the period $\tau_{stable}$. i.e for variable $X$:

$$X^* = 1\tau_{stable} \int_{\tau_{stable}} X dt \qquad (7)$$

Rearranging, we find the expression for $\phi$

$$\phi = W_{NSC}^* + K_m F_Q^*(T) W_{NSC}^* (\Pi_G C_v)^* \qquad (8)$$

By definition, the average NSC mass fraction is equal to $f_{NSC}$. Using this and equation (6 in the original manuscript), this becomes

$$\phi = 1 + a_{K_m} F_Q^*(T) (\Pi_G C_v)^* \qquad (9)$$

This means that to evaluate $\phi$, we require an estimate of average specific GPP over some stable period, and the average temperature during that period. If SUGAR is used at a single site these can be evaluated directly using GPP, biomass and temperature data if these are available. If these data are not available then the specific GPP can be approximated as the steady state carbon residency time, $\tau$ (e.g. in Carvalhais et al. 2004) and the temperature can found using global climatology data over the same period.

[Figure]

To evaluate $\alpha$, we re-write equation (4) as:

$$0 = \int \left( \Pi_N C_v - G C_v \right) dt \tag{10}$$

Again we divide by the integration period, $\tau_{stable}$, and assume this can be written as:

$$0 = \Pi_N^* C_v^* - G C_v^* \tag{11}$$

hence:

$$\Pi_N^* = G^* \tag{12}$$

Similarly using equation (4), we find

$$\Pi_G^* = U^* \tag{13}$$

Dividing equation (12) by equation (13) gives:

$$\alpha = CUE^* \tag{14}$$

where $CUE^* = \Pi_N^* \Pi_G^*$, is the time averaged carbon use efficiency of the non-stressed forest over the period $\tau_{stable}$.

Again this can be evaluated using data from a single site if available, or using more general estimates of CUE (e.g. Chambers et al. 2004, Gifford et al. 1995, Dewar 1998) if not.

2. **Coupled simulations**

When SUGAR is coupled to a LSM, the rate of change of structural biomass cannot be neglected, and we must consider the following equation.

$$dC_v dt = G - \Lambda \tag{15}$$

where $\Lambda$ represents all losses of structural carbon (i.e mortality $+$ litter-fall) and its form depends on which LSM is being used.

We make the further assumption that NSC is lost at a rate proportional to the loss of structural carbon. Equation (1 in the original manuscript) then becomes:

$$dC_{NSC}dt = \Pi_G - G - R_p - W_{NSC}\Lambda \qquad (16)$$

Substituting equations (15 and 16) into equation (1) gives

$$dW_{NSC}dt = \Pi_G C_v - (1 + W_{NSC})\,GC_v - R_p C_v \qquad (17)$$

To evaluate $\phi$ and $\alpha$ we then follow the same procedure as described above for uncoupled simulations. Doing so we find,

$$\phi = 1 + a_{K_m} F_Q^*(T)(1 + \alpha f_{NSC})\,(\Pi_G C_v)^* \qquad (18)$$

and

$$\alpha = CUE^*(1 + f_{NSC})(1 - f_{NSC}CUE^*) \qquad (19)$$

Again, these can be evaluated using GPP, NPP, biomass and temperature data from a single site, if available or using observed carbon residency time and carbon use efficiency at larger scales if not.

2. I felt a number of the plots were a little redundant. In fact, if you run their code, I'd say the time-series of NSC, perhaps with the addition of changing water availability, would be more insightful as to how the model works.

We appreciate your comment and note that the second reviewer also requested we include time-series of NSC. We attach a new figure of predicted NSC mass fraction in each plot from SUGAR (fig. 1). We also attach new figures that compare

time-series of predicted plant respiration (fig. 2) and PCE (fig. 3) from JULES and SUGAR. The reason for their initial omission was that besides the annual means of respiration and PCE presented in fig 6, there are no data for respiration, PCE or NSC to evaluate against the models against. We felt that since we could not evaluate any of these time-series against empirical data, it would not be worth including them in the manuscript. However, we accept your point that in terms of showing how the model works, these figures are much more useful than the figures provided.

3. I was a little bothered about how different the implementation of NSC actually was from LSMs that assume excess GPP goes into a labile pool, which is then used for growth/respiration? If I'm doing the authors a disservice here then I apologise, but perhaps a few more words outlining this distinction are required. I guess the bigger point I'm making here is that I was anticipating clear hypotheses about *how* and *when* such a labile pool would be used. I do not see these. For example, does the plant aim to maintain a minimum labile pool? What sets this? How big a pool does it accumulate? How would these things vary between PFTs? In the same way, what about the timing of utilisation? The authors spend the introduction sets up a clear link to water stress and this model as a plausible buffering mechanism. But the treatment of water stress in the manuscript is insufficient. It occurs to me that the authors are assuming that a plant will regulate GPP in the same way with and without a NSC pool (this is implied by the offline implementation). But does this make sense? The details are not given, but presumably in JULES water stress reduced the assimilation rate via reducing Vcmax. But if you have a NSC pool, would that imply a plant might be a little riskier? If it has some stockpile, why be quite so sensitive to water availability? I have nothing to support this line of thinking, but it seems pretty testable and logical (if only to me!).

In SUGAR, both respiration and growth depend on NSC pool size, meaning that only NSC is available to support respiration. This is supported by the recent work by Collalti

et al. 2019, that shows that respiration is neither strongly correlated with photosynthesis or total biomass but controlled more by labile carbon reserves. This is an important distinction between SUGAR and other representations of NSCs or labile carbon in other LSMs. As far as we are aware, many representations of NSC used in LSMs calculate respiration before considering the NSC pool meaning that when respiration demand exceeds total assimilate and reserve carbon, it is possible for either or both the growth flux or NSC pool to enter a negative state. This essentially means that the entire biomass pool is available to support respiration, which we know is not possible in plants

In this paper, we have focused on the original development of SUGAR and its first-order impact on fluxes (respiration and growth). Follow-up research, which will comprise the second and third chapter of my PhD, is ongoing to determine plant strategies for maintaining a minimum NSC pool size. We do not plan on including these further results in the paper but have responded to the reviewer questions below and will include some of this in the discussion section of our revised manuscript.

**How and when is the pool used?**
The pool is updated every model time step (which can range from half-hourly to daily timescales), so it is essentially always in use. Photosynthesis is only used to replenish the pool, while respiration and growth are always supported by the pool. The pool therefore represents both old NSC storage and more recent instantaneous assimilates. We will amend the current manuscript to make this clearer.

**How does the plant maintain NSC?**
The plants in SUGAR do not actively aim to maintain NSC. The NSC content is currently controlled purely by asynchrony between GPP and PCE . Forests are able to maintain NSC levels if specific photosynthetic rate $(\pi_G = \Pi_G C_v)$ remains constant. Specific photosynthesis is of course not constant and varies in response to changing climatic conditions which induces changes in the NSC pool. However, if

the perturbations in specific respiration cancel out over the period of, for example 1 year, then the NSC pool will be maintained over that year. If these perturbations do not cancel out, as is the case in the TFE simulation at Caxiuanã, then the NSC pool is depleted. In these simulations we are looking at the first order effect that this variation in NSC has on carbon fluxes and are neglecting further coupled effects that this may have on either vegetation dynamics or photosynthesis, which we felt were beyond the current scope of this current manuscript. Nonetheless, as pointed out by the reviewer, these are important processes that may significantly change the behaviour of plants within SUGAR, and hence are the focus of current ongoing research. Again we do not plan on including further results on this in this paper but below is a short discussion on these processes that we will include in our discussion section in a revised manuscript.

**Coupling the NSC pool to vegetation dynamics:**

1. **Litterfall and mortality driven by NSC depletion:**
   If the specific photosynthesis declines then this will cause a decrease in NSC mass fraction. We have explored the effect that this has on predictions of respiration and growth in this paper, but in theory this should also have an effect on vegetation dynamics. If NSC reserves remain depleted, and maintenance respiration is down-regulated as a result (as predicted by SUGAR) for a prolonged period, then we would expect this to be accompanied by a loss of biomass through either mortality or increased litter-fall, depending on how mortality is represented in the LSM being used. Representing this process is clearly a challenge given the scarcity of data directly showing carbon starvation induced mortality/biomass loss. SUGAR, however, can be used as a tool to test possible representations of these processes and drive and direct data collection.

2. **Active storage:**
   We are exploring the possibility of allowing the parameter $\phi$ to vary in time in response to either NSC content or specific photosynthetic rate, so that the plants

are able to actively regulate how NSC is used for respiration and growth depending on how much NSC or photosynthate is available. This may be necessary if specific photosynthesis declines significantly in models as forests grow (something we have found in JULES). Allowing $\phi$ to vary in time would also help to prevent the NSC pool from accumulating to unrealistic levels, although this is not a problem we have encountered too often and so we do include such a detailed discussion on this.

**Water stress and coupling the NSC pool to photosynthesis**

The only interaction with water availability comes through the already existing interaction with GPP in JULES. Coupling the NSC pool to GPP is again an important process but one we thought beyond the current scope of the model. Given recent developments in modelling stomatal closure and hydraulic damage (Eller 2019) this is something we hope to explore.

4. As I said above, the results are pretty convincing, but they are also indirect. There is nothing to support the model results being for the right mechanistic reasons. We have nothing that shows us the NSC timeseries (not shown) is supported experimentally from the through-fall experiment. We have nothing to say the respiration from this pool is supported experimentally. In both cases, I suspect a reader will anticipate such plots, I certainly did. Do such data exist? I have no idea if they do or not.

We regret that scaled-up NSC and plant respiration time-series do not exist from the throughfall experiment. This is in part, because scaling NSC measurements up to a whole plant and whole plot scale is difficult and has extremely large associated errors (Quentin 2015). Consequently, time-series of whole forest NSC stocks cannot be generated to a useful level of accuracy. This is similarly true for respiration data which is usually collected at an organ level. We have attached new figures of both NSC and

respiration time-series since as you rightly mentioned before they give a better idea of how the model works, however, unfortunately we are unable to include observations in these figures. In follow-up research we hope to evaluate SUGAR further at other Amazonian sites, where respiration data may be available through other measurement methods, i.e. via eddy-flux measurements.

5. Finally, the authors put forward an argument that the NSC model results at the throughfall experiment are limited by the poor representation of water stress on GPP in JULES. This is testable. All the authors have to do is make the GPP reduction less sensitive to water stress and plug these GPP values into their model. My suspicion is that the agreement with the obs may not improve, but I might be wrong. It would be worth testing this rather than speculating.

This is a very good suggestion and we attach two figures (figs. 4 & 5) which are the same as figures 5 and 6 in the original manuscript but with JULES simulations where the soil moisture stress has been reduced. The soil moisture stress in JULES is represented by multiplying photosynthesis by a piece-wise linear function of soil moisture, the so called '$\beta$ function'. Beta can be between 0 and 1, where 0 is complete soil moisture stress and 1 is no soil moisture stress. We have reduced the sensitivity of photosynthesis to soil moisture by simply defining a new beta function, $\beta' = min(\beta * 1.5, 1.0)$. This is clearly not a scientifically justifiable method, but for a quick and easy first look at how this would affect our results we think that it is acceptable. First looking at the control panel in both figures, you can see that this has not had a large effect on the model predictions, which is to be expected since there is already little soil moisture stress in the control simulations. With respect to the TFE plot, looking first at the bar graph of carbon fluxes, you can see that the reduction in soil moisture stress has improved predictions of PCE in both models, particularly in the start of the experiment. SUGAR in particular now accurately captures PCE in the first 4-5 years of the experiment where JULES does not. However, looking at predictions of NPP and respiration, as well as the accumulated NPP figure, you can see that while

SUGAR more accurately captures PCE, it is not capturing the allocation between NPP and respiration correctly. JULES interestingly now captures NPP accumulation in the TFE plot quite well. This is perhaps not too surprising since the assumptions made in SUGAR are relatively simple. In particular, this highlights that the assumption that growth and respiration have the same dependence on NSC may not hold for the TFE plot and that the forest is in some way prioritising respiration over growth. This in itself is an interesting result and we would be happy to include the above analysis and discussion in the revised manuscript.

Introduction ————

To be honest, I found the whole first paragraph completely unrelated to the focus of the paper. I think the paper would really benefit from a more relevant opening paragraph, entirely up to the authors what they do here, just a suggestion.

The first paragraph was written to provide a brief overview of the wider context of general land surface modelling. However, we agree that it is perhaps a step too far away from the focus of the paper and so in our revised manuscript we will remove the first paragraph and use what is currently the second paragraph which has a closer focus to the paper topic.

Pg 2, ln 24: "and so plants rely heavily on their NSC reserves". Are there any numbers to support this statement? How heavily? For how long?

There is unfortunately little data on an ecosystem scale that directly quantifies how much NSC is used during periods of drought, because, as stated above, coming up with whole plant estimations is still extremely complex and suffers from huge levels of uncertainty. The evidence of large discrepancy between utilisation and assimilation (Metcalfe et al. 2010, Doughty 2015 a,b) is given to imply that NSC must be relied upon during these periods, but we aren't aware of any ecosystem level measurements
that directly show NSC dependence. We will reword this to reflect the uncertainty, e.g.: "and so plants rely heavily on their NSC reserves" → "which implies that plants rely heavily on NSC reserves"

Pg 2, ln 41: what is the evidence for carbon starvation leading to mortality? My understanding is that it is essentially non-existent outside of a few potted experiments? See for example, Adams et al. A multi-species synthesis of physiological mechanisms in drought-induced tree mortality. I think this could be more carefully phrased.

This is a good point and it is true that there is not a large amount evidence that shows plants dying directly from carbon starvation. However, the point that we were trying to make here is that it is actually still not clear what the main driver of plant mortality is during drought. The theory tells us that both carbon starvation and hydraulic failure have the capacity to kill plants, but directly observing either process is extremely challenging.Nonetheless, what the literature actually shows is that plants are unlikely to die exclusively of carbon starvation or hydraulic failure and although one may trigger the path towards mortality, both carbon starvation and hydraulic failure are likely to be part of the mortality process (Sevanto et al. 2013). It is also likely that the two processes are not independent of each other since both carbon assimilation and water loss are controlled by stomatal conductance (Rowland et al. 2015). Consequently, capturing drought induced mortality will likely require representations of both hydraulics and carbohydrate storage. The other point to make is that most droughts do not actually lead to mortality and plants are resistant to most natural declines in water availability. The most important things to understand, therefore, are how plants are able to withstand drought and how they recover from them afterwards. We would argue that to understand these two processes requires an understanding of the theoretical threats that face plants during drought, and that while there is sparse evidence to show that carbon starvation is a significant killer during drought does not mean we should neglect it as a threat. Modelling NSC storage in LSMs may present

an opportunity to test the relationship between carbon starvation and hydraulic failure, in particular in conjunction with recent developments of stomatal modelling (Eller et al. 2018). It is probably fair to say that this is not really clear in our current manuscript and we have not really discussed the relationship between the two processes. We would be happy to include the above discussion in our revised manuscript to make our position clearer on this.

Pg 3, Second paragraph "Despite...". Obviously biased, and feel free to ignore...but I will draw your attention to "Mahmud et al. Inferring the effects of sink strength on plant carbon balance processes from experimental measurements. Biogeosciences.", which I think is a nice attempt to exploit experimental data to help mechanistically unpick the role of sink control. I highlight this paper because the focus was specifically to aid model development - "This can largely be attributed to a scarcity of ecosystem level data (NSC content and distribution) that can be used to parametrise and evaluate models for a range of species and climates that covers all plant functional types (PFTs) C3 BGD Interactive comment Printer-friendly version Discussion paper used in LSMs".

This is definitely a useful piece of work and thank you for bringing it to our attention. The revised manuscript will include this as an example of available NSC data for use in modelling efforts. It may also be useful in current and future work and developments of SUGAR where specific utilisation rates may vary depending on NSC build up or depletion as briefly discussed above.

Methods —— * Is there any supporting evidence for the assumptions in SUGAR? It's is fine if there isn't but it might be nice to cite some relevant literature if there is. For example, section 2.2...

This is a good suggestion and we note that the second reviewer also commented on the lack of discussion on some of the assumptions made in SUGAR. Below is a

discussion of the key assumptions in SUGAR and some justification behind them, which we will include in the model description to go alongside the model equations. The main assumptions in SUGAR are:

1. **GPP is all collected by a single pool of carbohydrate (sugars and starches are not distinguished) which is entirely readily available for respiration (R) and growth (G).**
   The is a simplification that we believe is necessary to represent NSC dynamics in large-scale land surface models like JULES, and it is a necessary assumption to keep the model simple and parameter sparse. There is obviously no evidence to support a single NSC pool that is readily available for use. In theory we could apply SUGAR to multiple organ tissues (eg. In Rastetter et al 1991), however, this would require representation of transport between the pools, which are difficult to represent and parametrise. Also many LSMs do not represent distinct wood tissues such as heartwood. Many LSMs, including JULES, split biomass allometrically between carbon pools and splitting the NSC pool in this way would not add any new dynamics to the model.

2. **G and R vary with temperature as a $Q_{10}$ functional with a q10 value of 2.0.**
   The Q10 function is a commonly used representation of the response of plant respiration and growth to temperature and a q10 value of 2.0 is also a standard value (Ryan 1991). Changing the temperature dependence in SUGAR would be a relatively easy procedure and so this could be explored in future work, however, for the purposes of this work we felt that using this more simple representation would be sufficient.

3. **G and R depend on NSC via a Michaelis-Menten function.**
   The Michaelis-Menten equation is a widely used description of enzyme kinetic which can be applied to both plant respiration and growth Thornley (1971).

4. **R and G have the same NSC dependence**

The assumption that maintenance respiration and growth share the same NSC dependence essentially reduces maintenance and growth respiration to one variable. Combining equations (4), (7) and (8), total plant respiration can be written as:

$$R_p = (R_{m_0} + 1 - Y_g Y_g G_0) F_Q(T) C_v C_{NSC} C_{NSC} + K_m C_v$$

which, using that $\phi = R_{m_0} + G_0 Y_g$ and equation (14), can be written as:

$$R_p = (1 - \alpha\alpha) G$$

It can therefore be interpreted that all respiration is associated with the growth of new structural material. This is explored in Thornley (2011). Using this assumption, Thornley is able to replicate the same results that are achieved using the more classical maintenance-growth respiration paradigm. For SUGAR this assumption is an important one that drastically simplifies the parameter estimation process, which was one of the main aims of developing the model. While we recognise that it is indeed a simplification of reality, we felt that the work in Thornley (2011), was sufficient justification given the benefit that using it provides.

In 2.5, it would be useful for the parameter ranges to be given to the reader? The section is titled parameter estimation but I've got no idea after reading it what values were used. I think a table with assumed parameters and/or ranges would be very useful for a reader who wished to repeat any of this. Currently, the only defined terms are the Q10 and Yg. For example, in the results: "All other parameters (Yg, aKm , q10) are kept constant at their default values (see model description)." Where was the value of aKm given?

We have updated this section to make the parameter evaluation process clearer.

We have also included parameter ranges where appropriate in the new parameter table above. With regards to $a_{K_m}$, we acknowledge that this not a well evaluated parameter, in the sense that it is not evaluated using empirical data. The default value of 0.5 was chosen since it gives numerically stable results (smaller values of $a_{K_m}$ can cause numerical instability with the commonly used minimum time-step of 1 hour). Larger values of $a_{K_m}$ also mean that the saturation effect that is provided by using michaelis-menten kinetics do not come into effect within a reasonable range of NSC concentrations. We accept that this is not a scientifically rigorous justification and so attach amended plots (figs. 6 & 7) from the Caxiuanã simulations with an in-built sensitivity study for $a_{K_m}$. The range of $a_{K_m}$ values tested is 0.1-2.0. These figures will replace figures 5 and 6 in the original manuscript.

I think the methods would benefit from a few sentences/paragraph explaining how the model works beyond the equations. Most of the introduction set up an interpretation of the use of NSC during periods of water stress but this theme is not returned to once in the methods. How does water stress interact here? It clearly isn't directly, but just comes about due to growth demand? What about the timescales of utilisation or storage increase? My reading of the model is that there is no specific hypothesis being tested here about increases in NSC. It is simply the difference between C uptake and utilisation. I don't really see that this goes beyond what many LSMs currently assume with respect to a excess carbon storage pool. I was expected a hypothesis about how plants might aim to maintain a storage pool, which I do not see. Equally, something about how they might prioritise a draw-down of this pool.

We refer back to the above discussion on NSC maintenance in point 3 of this response and to the discussion of model assumptions in the first point under methods——. We will include these discussions in the revised manuscript to give a clearer description of how the model works outside of the equations and the distinction we see from previous work.

Pg 6: "optimised so that annual GPP and NPP in the control forest agree with observations." What specifically was optimised here?

We used a previous configuration of JULES that had been parametrised using data from Caxiuanã but found that GPP was being underestimated relative to the control data from Metcalfe et al. 2010 and Da Costa et al. 2014) so we increased leaf nitrogen content, which increased predicted GPP.

Specifically we changed parameters:
vint from 7.21 to 12.0
vsl from 19.22 to 25.0
We also changed fdr from 0.01 to 0.0075 to correct carbon use efficiency in the control plot. The same parametrisation was then used in the TFE simulation.

Surely the SUGAR model was simple enough to just embed in JULES. Some further explanation is warranted here as to why this wasn't just done...

We are currently working on coupling SUGAR into JULES. The reason we did not do this in this paper is that it introduced some unexpected questions about how NSC interacts with processes including competition, mortality and land use change which are all modelled in the vegetation dynamics module in JULES (TRIFFID). There are also issues that relate to long term growth in JULES in which as the forest grows, specific photosynthesis declines and the forest is unable to maintain NSC concentration. We are looking at solving this by introducing an implicit active storage component to SUGAR by allowing $\phi$ to vary in time in response to either carbohydrate content or specific photosynthesis rate.

We recognise and accept that these are potentially very impactful processes

that may change the assumptions in SUGAR, however, ongoing work on this is suggesting that on the time-scales of the simulations in this paper, there will not be a significant change in the behaviour of SUGAR, and therefore the results of this work, beyond the realm of standard differences between coupled and uncoupled model simulations. Our main aim here was really to explore how SUGAR affects predictions of carbon fluxes (i.e respiration and growth) and we are aiming to look at vegetation dynamics (by coupling to TRIFFID) in future work.

Results ——
* As a general statement, I found it odd to start with the spatial interpretation rather than a site-level analysis. Doing it this way round is harder to see how the model is really working and to me (at least), it would make more sense to reverse the presentation of the results. Or alternatively, it would be useful to see a time-series extracted from Fig 2. For example, "This decline in seasonal variation is caused by an increase in dry season carbon expenditure and a decrease in the wet season carbon expenditure.". It would be nice to see this...

The aim of the spatial experiment was to demonstrate that modelling NSC has a significant impact on large scale ecosystem modelling, which is really the main purpose of SUGAR. The Caxiuanã experiments were then conducted to provide a more detailed evaluation, at a scale where data could be easily compared to the model output. Our main aim is to demonstrate that modelling NSC does not have a negligible effect on predictions of ecosystem carbon fluxes, rather than to improve simulations of the Caxiuanã drought experiment, which is why we have presented the simulations in this order. However, we accept that the spatial experiment does not clearly demonstrate how the model works so we attach a time-series of basin averaged simulated PCE along with the basin average driving GPP, that shows more clearly what SUGAR is doing to predictions of PCE (fig. 8).

I understand Fig 2 is a sensitivity experiment, but how are we meant to interpret whether the SUGAR model is improving / degrading growth predictions? I can see that increasing the fNSC dampens the variation, but is this dampening supported in any way? I wonder if Fig 3 is strictly necessary? It seems to be implied by Fig 2, I feel like you need one or the other. Perhaps it is a supplementary figure. I'd much rather a few time-series plots!

Unfortunately data that is sufficiently resolved to see this buffering effect at a basin scale doesn't exist, as far as we are aware. For this reason we originally only looked at the general effect that SUGAR has using statistical metrics rather than presenting time-series.

The implication of Fig 4 is that the respiration assumption in the model becomes more important as fNSC increases. How sensible is the assumed respiration eqn...this seems quite important.

The respiration equation is certainly not as detailed as many models but we feel that it captures the essential elements (i.e temperature and carbon availability). Please see the discussion on the key assumptions of SUGAR, above in point 1 of Methods in this response.

In the first paragraph of 4.2, it would be good to explain why JULES NPP and SUGAR NPP differ at all? If there is no water stress where does this difference come from? Do the models have different respiration assumptions? A shift in time-scale of growth?

SUGAR, at least in part, buffers any change in GPP (see new Caxiuanã PCE time-series plot). In the control simulations there is natural seasonal variation in GPP that is buffered by the NSC pool, which changes both respiration and growth relative to

JULES. This change is relatively small since the variation in GPP is small compared to the TFE simulation, but is sufficient to cause changes in the predictions of NPP.

Following the sensitivity experiment in Fig 2 onwards, my interpretation was that it was therefore not obvious how to parameterise the model. As such, I was expecting to see some form of uncertainty envelope around the SUGAR model line in Fig 5? How was SUGAR parametrised in this set of runs? I found this very unclear in my head at this point of the manuscript.

In the Caxiuanã simulations, SUGAR was parametrised using the first year of output data from JULES (i.e the year before the panels were put in), together with empirical data as described in the parameter table above. Parameters, $f_{NSC}$, $q_{10}$ and $Y_g$ were parameterised using empirical data or commonly used values as described above. Parameters $\alpha$ and $\phi$ were parametrised using the first year of JULES output. For example, $\alpha$ was found by taking the average CUE over this year. For $a_{K_m}$ we have since conducted a sensitivity study, since this was the least constrained parameter and our updated plots now show an uncertainty envelope based on allowing $a_{K_m}$ to vary between 0.1 and 2.0. Hopefully this process has been clarified by our discussion above, which we will include in the Methods section of the revised manuscript.

With Fig 5, arguably you don't need panel b, you could perhaps then include a respiration comparison between JULES and SUGAR? No idea what that looks like...

We will include both a comparison of respiration and PCE between SUGAR and JULES as well as the NSC time-series from SUGAR for each plot in the revised manuscript (see attached).

**References**

- Würth, M.K.R., Peláez-Riedl, S., Wright, S.J. et al.  Non-structural carbohydrate pools in a tropical forest.  Oecologia 143, 11–24 (2005). https://doi.org/10.1007/s00442-004-1773-2

- Ryan, M.G. (1991), Effects of Climate Change on Plant Respiration. Ecological Applications, 1: 157-167. doi:10.2307/1941808

- Thornley, J. H. M. and Johnson, I. R.: Plant and Crop Modelling. A Mathematical Approach to Plant and Crop Physiology, The Blackburn Press, 1990.

- Chambers, J.Q., Tribuzy, E.S., Toledo, L.C., Crispim, B.F., Higuchi, N., Santos, J.d., Araújo, A.C., Kruijt, B., Nobre, A.D. and Trumbore, S.E. (2004), RESPIRATION FROM A TROPICAL FOREST ECOSYSTEM: PARTITIONING OF SOURCES AND LOW CARBON USE EFFICIENCY. Ecological Applications, 14: 72-88. doi:10.1890/01-6012

- Gifford R.M. (1994) The global carbon cycle: a viewpoint on the missing sink. Australian Journal of Plant Physiology 21, 1– 15.

- Dewar, R.C., Medlyn, B.E. and McMurtrie, R.E. (1998), A mechanistic analysis of light and carbon use efficiencies.  Plant, Cell & Environment, 21: 573-588. doi:10.1046/j.1365-3040.1998.00311.x

- Carvalhais, N., Forkel, M., Khomik, M. et al.  Global covariation of carbon turnover times with climate in terrestrial ecosystems.  Nature 514, 213–217 (2014). https://doi.org/10.1038/nature13731

- Collalti, A, Tjoelker, MG, Hoch, G, et al.  Plant respiration: Controlled by photosynthesis or biomass?  Glob Change Biol.  2020; 26: 1739– 1753. https://doi.org/10.1111/gcb.14857

- Metcalfe, D.B., Meir, P., Aragão, L.E.O.C., Loboâ ĂŘdoâ ĂŘVale, R., Galbraith, D., Fisher, R.A., Chaves, M.M., Maroco, J.P., da Costa, A.C.L., de Almeida, S.S., Braga, A.P., Gonçalves, P.H.L., de Athaydes, J., da Costa, M., Portela, T.T.B., de Oliveira, A.A.R., Malhi, Y. and Williams, M. (2010), Shifts in plant respiration and carbon use efficiency at a largeâ ĂŘscale drought experiment in the eastern Amazon. New Phytologist, 187: 608-621. doi:10.1111/j.1469-8137.2010.03319.x

- Doughty, C. E., Metcalfe, D. B., Girardin, C. A. J., Amezquita, F. F., Durand, L., Huaraca Huasco, W., Silvaâ ĂŘEspejo, J. E., Araujoâ ĂŘMurakami, A., da Costa, M. C., da Costa, A. C. L., Rocha, W., Meir, P., Galbraith, D., and Malhi, Y. ( 2015a), Source and sink carbon dynamics and carbon allocation in the Amazon basin. Global Biogeochem. Cycles, 29, 645– 655. doi: 10.1002/2014GB005028.

- Doughty, C., Metcalfe, D., Girardin, C. et al. Drought impact on forest carbon dynamics and fluxes in Amazonia. Nature 519, 78–82 (2015b). https://doi.org/10.1038/nature14213

- SEVANTO, S., MCDOWELL, N.G., DICKMAN, L.T., PANGLE, R. and POCK-MAN, W.T. (2014), How do trees die?. Plant Cell Environ, 37: 153-161. doi:10.1111/pce.12141

- Rowland, L., da Costa, A., Galbraith, D. et al. Death from drought in tropical forests is triggered by hydraulics not carbon starvation. Nature 528, 119–122 (2015). https://doi.org/10.1038/nature15539

- Eller, C. B., Rowland, L., Oliveira, R. S., Bittencourt, P., Barros, F. V., da Costa, A., Meir, P., Friend, A. D., Mencuccini, M., Sitch, S., & Cox, P. (2018). Modelling tropical forest responses to drought and El Niño with a stomatal optimization model based on xylem hydraulics. Philosophical transactions of the Royal Society of London. Series B, Biological sciences, 373(1760), 20170315. https://doi.org/10.1098/rstb.2017.0315

- Edward B. Rastetter, Michael G. Ryan, Gaius R. Shaver, Jerry M. Melillo, Knute J. Nadelhoffer, John E. Hobbie, John D. Aber, A general biogeochemical model describing the responses of the C and N cycles in terrestrial ecosystems to changes in CO2, climate, and N deposition, Tree Physiology, Volume 9, Issue 1-2, July 1991, Pages 101–126, https://doi.org/10.1093/treephys/9.1-2.101

- Thornley, J. H. M.: Energy, Respiration, and Growth in Plants, Annals of Botany, 35, 721–728, https://doi.org/10.1093/oxfordjournals.aob.a084519, https://doi.org/10.1093/oxfordjournals.aob.a084519, 1971.

- Thornley, J. H. M.: Plant growth and respiration re-visited: maintenance respiration defined -it is an emergent property of, not a rate process within, the system - and why the respiration : photosynthesis ratio is conservative, Annals of Botany, 108, 1365–1380, https://doi.org/10.1093/aob/mcr238, https://doi.org/10.1093/aob/mcr238, 2011.

- da Costa, A. C. L., Metcalfe, D. B., Doughty, C. E., de Oliveira, A. A., Neto, G. F., da Costa, M. C., Silva Junior, J. d.  A., Aragão,L. E., Almeida, S., Galbraith, D. R., Rowland, L. M., Meir, P., and Malhi, Y.:  Ecosystem respiration and net primary productivity after 8–10 years of experimental through-fall reduction in an eastern Amazon forest, Plant Ecology & Diversity, 7, 7–24, https://doi.org/10.1080/17550874.2013.798366, https://doi.org/10.1080/17550874.2013.798366, 2014.

| Parameter | Units | Value (Cax) | Range | Description | Justification |
|---|---|---|---|---|---|
| $f_{NSC}$ | kgkg$^{-1}$ | 0.16 | 0.1-0.4 | Equilibrium NSC mass fraction | (Wurth et al 2005) |
| $q_{10}$ | | 2.0 | | Factor by which respiration and growth increase given a 10 degree warming | (Ryan 1991) |
| $Y_g$ | | 0.75 | | Growth conversion efficiency | (Thornley and Johnson 1990) |
| $a_{K_m}$ | | 0.1-2.0 | 0.1-2.0 | Relates the half saturation NSC mass fraction ($K_m$) with the equilibrium pool size ($f_{NSC}$) | Sensitivity study carried out in this study |
| $CUE^*$ | | 0.32 | 0.3-0.5 | Steady state carbon use efficiency | Used to evaluate $\alpha$. **Uncoupled run:** $\alpha = CUE^*$ **Coupled run:** $\alpha = CUE^*1 + f_{NSC}(1 - CUE^*)$. $CUE^*$ is usually between 0.3-0.5 for a tropical forest (Chambers et al 2004, Gifford 1995, Dewar 1998) *see parameter evaluation for derivation* |
| $\tau$ | yr | 5.15 | 1.18 - 6.49 | Steady state vegetation carbon residency time | Used to evaluate $\phi$. **Uncoupled run:** $\phi = 1 + a_{K_m}\tau F_Q^*(T)$ **Coupled run:** $\phi = 1 + a_{K_m}\tau F_Q^*(T)(1 + \alpha f_{NSC})$. e.g. (Carvalhais 2004) *see parameter evaluation for derivation* |

$f_{NSC}$ the is fraction of NSC relative to total structural carbon and so estimates of NSC as a fraction of total dry mass should be adjusted to account for non-carbon biomass.

**Table 1.** Parameters in SUGAR

---

## Author Comment (AC2) · 26 Mar 2020

[bg]copernicus xcolor

[Figure]

Response to Thomas Pugh

1. Reflecting the lack of observations of NSC stocks and dynamics, the model is designed to be parametrised based on more commonly-available measurements such as biomass and GPP. This is an elegant idea, however some fairly substantial assumptions are made to achieve it, and those assumptions are not well explored here. One key assumption, made in Eq. 8, is that the NSC dependence of maintenance respiration is the same as that for plant growth. The implicit hypothesis is that respiration and growth are equally prioritised under resource limitation. As far as I am aware, there is no clear evidence to support this and it is just as likely that the plant should down regulate growth to maintain respiration-related functions. No reasoning or citations are presented to support this key choice. In my opinion a thorough discussion is necessary.

The assumption that maintenance respiration and growth share the same NSC dependence essentially reduces maintenance and growth respiration to one variable. Combining equations (4), (7) and (8), total plant respiration can be written as:

$$R_p = \left(R_{m_0} + 1 - Y_g Y_g G_0\right) F_Q(T) C_v C_{NSC} C_{NSC} + K_m C_v$$

which, using that $\phi = R_{m_0} + G_0 Y_g$ and equation (14), can be written as:

$$R_p = (1 - \alpha\alpha)\, G$$

It can therefore be interpreted that all respiration is associated with the growth of new structural material. This is explored in Thornley (2011). Using this assumption, Thornley is able to replicate the same results that are achieved using the more classical maintenance-growth respiration paradigm. For SUGAR this assumption is an important one that drastically simplifies the parameter estimation process, which was one of the main aims of developing the model. While we recognise that it is

indeed a simplification of reality, we felt that the work in Thornley (2011), was sufficient justification given the benefit that using it provides. We will include this discussion in a revised manuscript along with a further discussion of the other assumptions in SUGAR (requested by other reviewer), specifically:

1. **GPP is all collected by a single pool of carbohydrate (sugars and starches are not distinguished) which is entirely readily available for respiration (R) and growth (G).**
The is a simplification that we believe is necessary to represent NSC dynamics in large-scale land surface models like JULES, and it is a necessary assumption to keep the model simple and parameter sparse. There is obviously no evidence to support a single NSC pool that is readily available for use. In theory we could apply SUGAR to multiple organ tissues (eg. Rastetter et al 1991), however, this would require representation of transport between the pools, which are difficult to represent and parametrise. Also many DGVMs do not represent distinct wood tissues such as heartwood. Many DGVMs, including JULES, split biomass allometrically between carbon pools and splitting the NSC pool in this way would not add any new dynamics to the model.

2. **G and R vary with temperature as a $Q_{10}$ functional with a q10 value of 2.0.**
The Q10 function is a commonly used representation of the response of plant respiration and growth to temperature and a q10 value of 2.0 is also a standard value (Ryan 1991). Changing the temperature dependence in SUGAR would be a relatively easy procedure and so this could be explored in future work, however, for the purposes of this work we felt that using this more simple representation would be sufficient.

3. **G and R depend on NSC via a Michaelis-Menten function**
The Michaelis-Menten equation is a widely used description of enzyme kinetic re-

actions that can be applied to both plant growth and respiration Thornley (1971).

2. I am also a bit curious about the assumption implied in Eq. 3 that NSC in the heartwood is available to trees to use. Is it posited that there are mechanisms that trees can use to extract this NSC from dead wood, in which case citations are needed, or is this simply an assumption to enable use of the widely available total biomass information? The latter would seem perfectly reasonable to me, but then does seem to warrant a short discussion about the fraction of total NSC that is typically found in heartwood (e.g. <15%, Richardson et al., 2015).

The limitation here is that most LSMs like JULES do not represent heartwood versus sapwood. Consequently we are forced to assume that the entire pool is readily available to the forest/plant. Many LSMs including JULES, split biomass allometrically between carbon pools and splitting the NSC pool in this way would not add any new dynamics to the model. We accept that this is again a simplification of a reality, however it is a necessary assumption to keep the model simple and parameter sparse. In theory we could apply SUGAR to multiple organ tissues (eg. In Rastetter et al 1991), however, this would require representation of transport between the pools, which are difficult to represent and evaluate.

3. Similarly, there has been clear effort to minimise the number of parameters, but choices of parameter values are not well justified in the text. Parameters aKm and q10 are assigned default values with no justification given for the choice. In particular, aKm appears to be central to the results. If clear, strong, justification for the choices cannot be given, then the sensitivity of the results to these parameters should be included in the tests. The parameter fNSC is appropriately treated as a range, although I would have been inclined to set upper limit of the range a bit above that actually observed in a tropical forest to explore the parameter space a bit, but fair enough!

$f_{NSC}$: It is a good suggestion to extend the range of tested $f_{NSC}$ values. This was not done originally as there didn't seem to be a clear or scientific way to pick a higher value. Additionally the buffering effect of the model seems to saturate at larger values of $f_{NSC}$ and we felt that the transition from no NSC to 'some' NSC was captured fairly well with the given range. Nonetheless, attached is are updated plots (figs. 1, 2 & 3) with $f_{NSC} = 0.16$, chosen as double the original maximum tested value.

$q_{10}$ = 2.0: This is a relatively standard value for Q10 temperature relationships (Ryan 1991) and is commonly used in DGVMs. Changing the temperature dependence in SUGAR would be a relatively easy procedure and so this could be explored in future work, however, for the purposes of this work we felt that using a standard value of 2.0 would be acceptable.

$a_{K_m}$ is admittedly a not well evaluated parameter, in the sense that it is not evaluated using empirical data. The default value of 0.5 was chosen since it gives numerically stable results (smaller values of $a_{K_m}$ can cause numerical instability with the commonly used minimum time-step of 1 hour). Larger values of $a_{K_m}$ also mean that the saturation effect that is provided by using michaelis-menten kinetics do not come into effect within a reasonable range of NSC concentrations. We accept that this is not a scientifically rigorous justification and so attach amended plots from the Caxiuanã (figs. 4 & 5) simulations with an in-built sensitivity study for $a_{K_m}$. The range of $a_{K_m}$ values tested is 0.1-2.0. These figures will replace figures 5 and 6 in the original manuscript.

Pg. 4, lines 11-12. So this means that SUGAR assumes that trees actively allocate to storage to maintain a certain store size. Perhaps acknowledge this decision explicitly here, given the active debate on this (which was introduced in the introduction)?

This section may not be worded particularly well. Currently SUGAR assumes no active

storage and NSC content is regulated passively by the asynchrony between GPP and U. The assumption that we are making here is that during non-stressed conditions plants do not rely significantly on their NSC reserves so that over the course of a period of one year, the mass fraction of NSC ($C_{NSC}C_v$) remains constant. We then define the parameter $f_{NSC}$ to equal this constant value, which we evaluate using observed NSC mass fractions (Wurth et al 2005). This is then used to determine NSC turnover rate ($\phi$). If over a long period of time these perturbations do not average to zero, as in the TFE simulations at Caxiuanã then the NSC pool will either accumulate or deplete. In this study we look primarily at the first-order effect that this has on ecosystem carbon fluxes (respiration and growth) and have neglected the effects on vegetation dynamics. This is clearly an important process that is the focus of ongoing and future work, but we felt was beyond the scope of this paper. Below is a further discussion on some of our ongoing work.

We will change the description here to make this clearer. We also include a more detailed discussion on this assumption below which we will include in the derivation of parameter evaluation section in the supplementary materials.

Further discussion:
The NSC pool varies in response to both changes in temperature (via Q10 function $F_Q$) and changes in GPP ($\Pi_G$). More usefully, the NSC mass fraction ($C_{NSC}C_v$) varies in response to changes in temperature and specific photosynthesis ($\pi_G = \Pi_G C_v$). If we write specific photosynthesis as the sum of its average value and variations caused by changes in environment: $\pi_G = \bar{\pi}_G + \pi'_G$, and similarly $F_Q = \bar{F}_Q + F'_Q$ then the NSC mass fraction will deviate from it's equilibrium value if $\pi'_G$ or $F'_Q$ are non zero. This of course happens all the time since GPP and temperature vary on a sub-daily time-scale. However, if over some longer time period these perturbations average to zero then the NSC mass fraction will remain constant over that period. We assume that this is the case in forests under non-stressed conditions that are generally

observed to have relatively stable NSC pools. We then set $f_{NSC}$ equal to observed NSC mass fractions in such forests.

Coupling NSC pool to vegetation dynamics
The two processes we are looking at are:

1. **Litterfall and mortality driven by NSC depletion:**
   If the specific photosynthesis declines then this will cause a decrease in NSC mass fraction. This should then be accompanied by a loss of structural biomass through either a litter-fall or mortality term, that restores NSC mass fraction to its equilibrium value. We felt that this interaction was beyond the scope of this paper where we were looking just at carbon fluxes (R and G) but are exploring it as part of current research.

2. **Active storage:**
   We are exploring the possibility of allowing the parameter $\phi$ to vary in time in response to either NSC content or specific photosynthetic rate, so that the plants are able to actively regulate how NSC is used for respiration and growth depending on how much NSC or photosynthate is available. This may be necessary if specific photosynthesis declines significantly in models as forests grow (something we have found in JULES). Allowing $\phi$ to vary in time would also help to prevent the NSC pool from accumulating to unrealistic levels, although this is not a problem we have encountered too often and so we do include such a detailed discussion on this.

Pg. 7, line 8. Is this above-ground NPP or total NPP? If the former, how is the output from SUGAR being adjusted to compensate for this in the evaluation?

The observations here show total NPP except for the root increment component, as this data was not available from the site. The model outputs from SUGAR and JULES

are both total NPP and so we have made a correction in the attached figures (fig. 5). For the JULES simulation we have subtracted simulated root increment ($drootC$) from simulated NPP. Since SUGAR does not distinguish between root, stem, leaf etc we take the fraction $\frac{drootC}{NPP_{total}}$ from JULES and multiply by total simulated NPP in SUGAR. This is then taken from the simulated NPP in SUGAR. Simulated root C increment in JULES is negligible relative to total simulated NPP and so this correction does not qualitatively change our results (see updated figure), but we will update our results section in the revised manuscript with the corrected numbers.

Pg. 7, line 16. How is the CV calculated? Annual mean of daily (or monthly) values, followed by taking the mean across the simulated years? Or directly over the whole dataset? I'm trying to understand if this is showing intra- or interannual variability.

CV is calculated as standard deviation/mean so for each gridbox, the standard deviation and mean of simulated PCE are calculated over the entire simulation period which is outputted on a monthly time-step. The mean CV quoted is then the spatial mean across the basin. It is a metric of how variable PCE is over the entire period. Both inter and intra annual variability should be captured by this.

Pg. 8, line 31. Doesn't this imply that the NSC store has been underestimated? Perhaps worth exploring how much storage would be required to maintain respiration and growth and how this is affected by parameters like aKm? This could provide a useful hypothesis for future investigation.

This is essentially what we have tried to explore in the discussion paragraph starting Pg. 10, line 10. Rather than explicitly using SUGAR though, we make an inversion using the observed PCE and predicted GPP by JULES to determine the deficit of carbon required to support growth and respiration.

Pg. 9, lines 42-44. Maybe a bit of over interpretation of small differences here? Overall JULES and SUGAR seem equally good for the control plots in Figs. 5 and 6.

It is true that the models do perform very similarly on the control plot, which is really the main positive result of the control simulations, since JULES is already able to capture control fluxes quite accurately. However, the models do perform differently on shorter time-scales, which we think is still an important result that follows on from the basin level simulation results in section 4.1. Admittedly the original figures (5&6) don't really show this very well. The differences are better illustrated by the attached plot of monthly PCE in SUGAR and JULES (fig. 6) which we will include the revised manuscript. This figure shows how PCE is buffered against the natural seasonal variation in GPP that the control forest experiences in JULES. The annual averages of PCE within JULES and SUGAR are very similar on the control plot, but as can be seen in this figure, the seasonal variation is different.

Pg. 9, line 51. Whilst the down-regulation of respiration in response to depleting NSC may help buffer NPP, it clearly doesn't improve the simulation of respiration, which is more strongly underestimated in SUGAR than in JULES from 2005 onwards. I think this discussion needs to reflect that whilst SUGAR provides a very useful representation of a process we are confident is important, including this process does not by itself radically improve the overall carbon flux simulation for the drought experiment investigated.

This is a fair comment and we will include this in our discussion. While SUGAR doesn't improve respiration predictions, it does improve predictions of total PCE for the first 5-6 years of the experiment. The reason that the prediction of respiration is not improved relative to JULES is related to the allocation to growth and respiration in SUGAR. We will shift the discussion of our results from NPP and respiration to the prediction of total PCE and also include more of a discussion on allocation of C between G and R. This will link to the discussion above on the validity of the assumption that R and G

[Figure]

have the same NSC dependence.

Fig. 5. It would be helpful to also see time-series of respiration and NSC storage to allow a full and balanced interpretation.

We agree that these figures would be useful and note that this was also requested by the first reviewer. The reason for their initial omission was that besides the annual respiration presented in fig 6, there are no data for R and NSC to evaluate these plots. However, as you say they still provide useful information about how the model works relative to JULES so please see the attached figures (7 & 8) which we will include in our revised manuscript.

**References**

- Thornley, J. H. M.: Plant growth and respiration re-visited: maintenance respiration defined -it is an emergent property of, not a rate process within, the system - and why the respiration : photosynthesis ratio is conservative, Annals of Botany, 108, 1365–1380, https://doi.org/10.1093/aob/mcr238, https://doi.org/10.1093/aob/mcr238, 2011.

- Edward B. Rastetter, Michael G. Ryan, Gaius R. Shaver, Jerry M. Melillo, Knute J. Nadelhoffer, John E. Hobbie, John D. Aber, A general biogeochemical model describing the responses of the C and N cycles in terrestrial ecosystems to changes in CO2, climate, and N deposition, Tree Physiology, Volume 9, Issue 1-2, July 1991, Pages 101–126, https://doi.org/10.1093/treephys/9.1-2.101

- Ryan, M.G. (1991), Effects of Climate Change on Plant Respiration. Ecological Applications, 1: 157-167. doi:10.2307/1941808

- Thornley, J. H. M.: Energy, Respiration, and Growth in Plants, Annals

of Botany, 35, 721–728, https://doi.org/10.1093/oxfordjournals.aob.a084519, https://doi.org/10.1093/oxfordjournals.aob.a084519, 1971.

- Würth, M.K.R., Peláez-Riedl, S., Wright, S.J. et al. Non-structural carbohydrate pools in a tropical forest. Oecologia 143, 11–24 (2005). https://doi.org/10.1007/s00442-004-1773-2

---

## Author Response (AR1)

We would like to thank both reviewers for their detailed and constructive comments and feedback. We present here the amendments that have been made in our revised manuscript along with the relevant discussion from our first response.

Response to reviewer 1

I felt the details on how the model was parametrised could be improved. I strongly felt the paper needs a table with parameters used, which would make this paper repeatable.

This is a good suggestion, and we agree that the description of model parametrisation, in general, is not sufficiently clear. We have added a table (Table 3. page 37) to show all the parameters in SUGAR, their ranges, and values used in the Caxiuanã simulations where applicable. We have also updated the parameter evaluation section (starting page 5 line 26) so that it more clearly outlines how each parameter should be evaluated. We have added a derivation of the expressions given for $\phi$ and $\alpha$ to the Appendix (starting page 13 line 3).

Finally we have also updated the methods section to more clearly outline how the parameters are evaluated in both experiments in the paper (Page 7 Line 22 and Page 8 Line 20). 2. I felt a number of the plots were a little redundant. In fact, if you run their code, I'd say the time-series of NSC, perhaps with the addition of changing water availability, would be more insightful as to how the model works.

We appreciate your comment and note that the second reviewer also requested we include time-series of NSC. We have included a new figure of predicted NSC mass fraction at Caxiuanã in each plot from SUGAR (Figure 7. Page 28). We have also added new figures that compare time-series of predicted plant respiration and PCE from JULES and SUGAR (Figures A3 & A4, Pages 31 & 32). The reason for their initial omission was that besides the annual means of respiration and PCE presented in Fig. 6, there are no data for respiration, PCE or NSC to evaluate against the models against. We felt that since we could not evaluate any of these time-series against empirical data, it would not be worth including them in the manuscript. However, we accept your point that in terms of showing how the model works, these figures may be useful. Since we are unable to evaluate these model outputs and to limit the number of main paper figures we have added these as appendix figures.

3. I was a little bothered about how different the implementation of NSC actually was from LSMs that assume excess GPP goes into a labile pool, which is then used for growth/respiration? If I'm doing the authors a disservice here then I apologise, but perhaps a few more words outlining this distinction are required. I guess the bigger point I'm making here is that I was anticipating clear hypotheses about *how* and *when* such a labile pool would be used. I do not see these. For example, does the plant aim to maintain a minimum labile pool? What sets this? How big a pool does it accumulate? How would these things vary between PFTs? In the same way, what about the timing of utilisation? The authors spend the introduction sets up a clear link to water stress and this model as a plausible buffering mechanism. But the treatment of water stress in the manuscript is insufficient. It occurs to me that the authors are assuming that a plant will regulate GPP in the same way with and without a NSC pool (this is implied by the offline implementation). But does this make sense? The details are not given, but presumably in JULES water stress reduced the assimilation rate via reducing Vcmax. But if you have a NSC pool, would that imply a plant might be a little riskier? If it has some stockpile, why be quite so sensitive to water availability? I have nothing to support this line of thinking, but it seems pretty testable and logical (if only to me!).

In SUGAR, both respiration and growth depend on NSC pool size, meaning that only NSC is available to support respiration. This is supported by the recent work by Collalti et al. 2019, that shows that respiration is neither strongly correlated with photosynthesis or total biomass but controlled more by labile carbon reserves. This is an important distinction between SUGAR and other representations of NSCs or labile carbon in other LSMs. As far as we are aware, many representations of NSC used in LSMs calculate respiration before considering the NSC pool meaning that when respiration demand exceeds total assimilate and reserve carbon, it is possible for either or both the growth flux or NSC pool to enter a negative state. This essentially means that the entire biomass pool is available to support respiration, which we know is not possible in plants

In this paper, we have focused on the original development of SUGAR and its first-order impact on fluxes (respiration and growth). Follow-up research, which will comprise the second and third chapter of my PhD, is ongoing to determine plant strategies for maintaining a minimum NSC pool size. We do not plan on including these further results in the paper but have updated the model description section to give a stronger description of the model outside of the equations (Page 4 Lines 2-14).

4. As I said above, the results are pretty convincing, but they are also indirect. There is nothing to support the model

results being for the right mechanistic reasons. We have nothing that shows us the NSC timeseries (not shown) is supported experimentally from the through-fall experiment. We have nothing to say the respiration from this pool is supported experimentally. In both cases, I suspect a reader will anticipate such plots, I certainly did. Do such data exist? I have no idea if they do or not.

We regret that scaled-up NSC and plant respiration time-series do not exist from the throughfall experiment. This is in part, because scaling NSC measurements up to a whole plant and whole plot scale is difficult and has extremely large associated errors (Quentin 2015). Consequently, time-series of whole forest NSC stocks cannot be generated to a useful level of accuracy. This is similarly true for respiration data which is usually collected at an organ level. We have attached new figures of both
10  NSC and respiration time-series since as you rightly mentioned before they give a better idea of how the model works, however, unfortunately we are unable to include observations in these figures. In follow-up research we hope to evaluate SUGAR further at other Amazonian sites, where respiration data may be available through other measurement methods, i.e. via eddy-flux measurements.

15  5. Finally, the authors put forward an argument that the NSC model results at the throughfall experiment are limited by the poor representation of water stress on GPP in JULES. This is testable. All the authors have to do is make the GPP reduction less sensitive to water stress and plug these GPP values into their model. My suspicion is that the agreement with the obs may not improve, but I might be wrong. It would be worth testing this rather than speculating.

This is a good suggestion and we have added two figures (Figures A6 & A5, Pages 33 & 34) which are the same as figures 5
20  and 6 in the original manuscript but with JULES simulations where the soil moisture stress has been reduced by 50%. The soil moisture stress in JULES is represented by multiplying photosynthesis by a piece-wise linear function of soil moisture, the so called '$\beta$ function'. Beta can be between 0 and 1, where 0 is complete soil moisture stress and 1 is no soil moisture stress. We have reduced the sensitivity of photosynthesis to soil moisture by simply defining a new beta function, $\beta' = min(\beta * 1.5, 1.0)$. This is clearly not a scientifically justifiable method, but for a quick and easy first look at how this would affect our results we
25  think that it is acceptable. We have included a short sentence of discussion on this on page 12 lines 10-12.

Introduction ————

To be honest, I found the whole first paragraph completely unrelated to the focus of the paper. I think the paper would really benefit from a more relevant opening paragraph, entirely up to the authors what they do here, just a suggestion.

The first paragraph was written to provide a brief overview of the wider context of general land surface modelling. However, we agree that it is perhaps a step too far away from the focus of the paper. We have removed the first paragraph and reworked the second paragraph to create a more relevant first paragraph (Page 1 lines 32-33 and Page 2 lines 1-13).

35  Pg 2, ln 24: "and so plants rely heavily on their NSC reserves". Are there any numbers to support this statement? How heavily? For how long?

There is unfortunately little data on an ecosystem scale that directly quantifies how much NSC is used during periods of drought, because, as stated above, coming up with whole plant estimations is still extremely complex and suffers from huge
40  levels of uncertainty. The evidence of large discrepancy between utilisation and assimilation (Metcalfe et al. 2010, Doughty 2015 a,b) is given to imply that NSC must be relied upon during these periods, but we aren't aware of any ecosystem level measurements that directly show NSC dependence. We have reworded this to reflect the uncertainty (Page 2 Line 35)).

Pg 2, ln 41: what is the evidence for carbon starvation leading to mortality? My understanding is that it is essentially
45  non-existent outside of a few potted experiments? See for example, Adams et al. A multi-species synthesis of physiological mechanisms in drought-induced tree mortality. I think this could be more carefully phrased.

This is a good point and it is true that there is not a large amount evidence that shows plants dying directly from carbon starvation. However, the point that we were trying to make here is that it is actually still not clear what the main driver of
50  plant mortality is during drought. The theory tells us that both carbon starvation and hydraulic failure have the capacity to kill plants, but directly observing either process is extremely challenging.Nonetheless, what the literature actually shows is that plants are unlikely to die exclusively of carbon starvation or hydraulic failure and although one may trigger the path towards mortality, both carbon starvation and hydraulic failure are likely to be part of the mortality process (Sevanto et al. 2013). It is also likely that the two processes are not independent of each other since both carbon assimilation and water loss
55  are controlled by stomatal conductance (Rowland et al. 2015). Consequently, capturing drought induced mortality will likely

require representations of both hydraulics and carbohydrate storage. The other point to make is that most droughts do not actually lead to mortality and plants are resistant to most natural declines in water availability. The most important things to understand, therefore, are how plants are able to withstand drought and how they recover from them afterwards. We have added some short discussion on this to the introduction (Page 2 lines 38-42).

Pg 3, Second paragraph "Despite...". Obviously biased, and feel free to ignore...but I will draw your attention to "Mahmud et al. Inferring the effects of sink strength on plant carbon balance processes from experimental measurements. Biogeosciences.", which I think is a nice attempt to exploit experimental data to help mechanistically unpick the role of sink control. I highlight this paper because the focus was specifically to aid model development - "This can largely be attributed to a scarcity of ecosystem level data (NSC content and distribution) that can be used to parametrise and evaluate models for a range of species and climates that covers all plant functional types (PFTs) C3 BGD Interactive comment Printer-friendly version Discussion paper used in LSMs".

This is definitely a useful piece of work and thank you for bringing it to our attention. The revised manuscript now includes this as an example of available NSC data for use in modelling efforts (Page 3 Line 27).

Methods —— * Is there any supporting evidence for the assumptions in SUGAR? It's is fine if there isn't but it might be nice to cite some relevant literature if there is. For example, section 2.2...

This is a good suggestion and we note that the second reviewer also commented on the lack of discussion on some of the assumptions made in SUGAR. We have included some discussion on this in the methods section (Page 4 lines 2-14), the new parameter evaluation section (Page 5 line 26 - Page 7 line 13) and in the discussion section (Page 11 lines 43-55) where briefly discuss some of the caveats of some of these assumptions.

In 2.5, it would be useful for the parameter ranges to be given to the reader? The section is titled parameter estimation but I've got no idea after reading it what values were used. I think a table with assumed parameters and/or ranges would be very useful for a reader who wished to repeat any of this. Currently, the only defined terms are the Q10 and Yg. For example, in the results: "All other parameters (Yg, aKm , q10) are kept constant at their default values (see model description)." Where was the value of aKm given?

We agree and as partially described above have updated the parameter evaluation section (Page 5 line 26) and included parameter ranges where appropriate in the new parameter table (Page 36). With regards to $a_{K_m}$, we acknowledge that this is not a well evaluated parameter, in the sense that it is not evaluated using empirical data. The default value of 0.5 was chosen since it gives numerically stable results (smaller values of $a_{K_m}$ can cause numerical instability with the commonly used minimum time-step of 1 hour) and sensible NSC mass fractions. We accept that this is not a scientifically rigorous justification and so have amended the Caxiuanã simulations and conducted a sensitivity study for $a_{K_m}$. The range of $a_{K_m}$ values tested is 0.1-2.0. These figures (Figs. 5 & 6) have replaced the original figures from Caxiuan a (Figs. 5 & 6 of original manuscript).

I think the methods would benefit from a few sentences/paragraph explaining how the model works beyond the equations. Most of the introduction set up an interpretation of the use of NSC during periods of water stress but this theme is not returned to once in the methods. How does water stress interact here? It clearly isn't directly, but just comes about due to growth demand? What about the timescales of utilisation or storage increase? My reading of the model is that there is no specific hypothesis being tested here about increases in NSC. It is simply the difference between C uptake and utilisation. I don't really see that this goes beyond what many LSMs currently assume with respect to a excess carbon storage pool. I was expected a hypothesis about how plants might aim to maintain a storage pool, which I do not see. Equally, something about how they might prioritise a draw-down of this pool.

We refer back to point 3 of this response and to the updated paragraph at the start of the model description (Page 4 lines 2-14).

Pg 6: "optimised so that annual GPP and NPP in the control forest agree with observations." What specifically was optimised here?

We used a previous configuration of JULES that had been parametrised using data from Caxiuanã but found that GPP was being underestimated relative to the control data from Metcalfe et al. 2010 and Da Costa et al. 2014) so we increased effective leaf nitrogen content, which increased predicted GPP.

Specifically we changed parameters:

vint from 7.21 to 12.0

vsl from 19.22 to 25.0

We also changed fdr from 0.01 to 0.0075 to correct carbon use efficiency in the control plot. The same parametrisation was then used in the TFE simulation. We have attached these details in a separate word document as supplementary information.

Surely the SUGAR model was simple enough to just embed in JULES. Some further explanation is warranted here as to why this wasn't just done...

We are currently working on coupling SUGAR into JULES. The reason we did not do this in this paper is that it introduced some unexpected questions about how NSC interacts with processes including competition, mortality and land use change which are all modelled in the vegetation dynamics module in JULES (TRIFFID). There are also issues that relate to long term growth in JULES in which as the forest grows, specific photosynthesis declines and the forest is unable to maintain NSC concentration. We are looking at solving this by introducing an implicit active storage component to SUGAR by allowing $\phi$ to vary in time in response to either carbohydrate content or specific photosynthesis rate.

We recognise and accept that these are potentially very impactful processes that may change the assumptions in SUGAR, however, ongoing work on this is suggesting that on the time-scales of the simulations in this paper, there will not be a significant change in the behaviour of SUGAR, and therefore the results of this work, beyond the realm of standard differences between coupled and uncoupled model simulations. Our main aim here was really to explore how SUGAR affects predictions of carbon fluxes (i.e respiration and growth) and we are aiming to look at vegetation dynamics (by coupling to TRIFFID) in future work.

Results ——

As a general statement, I found it odd to start with the spatial interpretation rather than a site-level analysis. Doing it this way round is harder to see how the model is really working and to me (at least), it would make more sense to reverse the presentation of the results. Or alternatively, it would be useful to see a time-series extracted from Fig 2. For example, "This decline in seasonal variation is caused by an increase in dry season carbon expenditure and a decrease in the wet season carbon expenditure.". It would be nice to see this...

The aim of the spatial experiment was to demonstrate that modelling NSC has a significant impact on large scale ecosystem modelling, which is really the main purpose of SUGAR. The Caxiuanã experiments were then conducted to provide a more detailed evaluation, at a scale where data could be easily compared to the model output. Our main aim is to demonstrate that modelling NSC does not have a negligible effect on predictions of ecosystem carbon fluxes, rather than to improve simulations of the Caxiuanã drought experiment, which is why we have presented the simulations in this order. However, we accept that the spatial experiment does not clearly demonstrate how the model works. We have added a new figure (Figure 2 Page 23) which shows a time-series of basin averaged simulated PCE along with the basin average driving GPP. To restrict the number of main paper figures, we have also moved Figure 2 from the original manuscript to the appendix (Figure A1 page 29).

I understand Fig 2 is a sensitivity experiment, but how are we meant to interpret whether the SUGAR model is improving / degrading growth predictions? I can see that increasing the fNSC dampens the variation, but is this dampening supported in any way? I wonder if Fig 3 is strictly necessary? It seems to be implied by Fig 2, I feel like you need one or the other. Perhaps it is a supplementary figure. I'd much rather a few time-series plots!

Unfortunately data that is sufficiently resolved to see this buffering effect at a basin scale doesn't exist, as far as we are aware. For this reason we originally only looked at the general effect that SUGAR has using statistical metrics rather than presenting time-series.

The implication of Fig 4 is that the respiration assumption in the model becomes more important as fNSC increases. How sensible is the assumed respiration eqn...this seems quite important.

The respiration equation is certainly not as detailed as many models but we feel that it captures the essential elements (i.e temperature and carbon availability). Please see the discussion on the key assumptions of SUGAR, above in point 1 of Methods in this response.

In the first paragraph of 4.2, it would be good to explain why JULES NPP and SUGAR NPP differ at all? If there is no water stress where does this difference come from? Do the models have different respiration assumptions? A shift in time-scale of growth?

SUGAR, at least in part, buffers any change in GPP (see new Caxiuanã PCE time-series plot). In the control simulations there is natural seasonal variation in GPP that is buffered by the NSC pool, which changes both respiration and growth relative to JULES. This change is relatively small since the variation in GPP is small compared to the TFE simulation, but is sufficient to cause changes in the predictions of NPP.

Following the sensitivity experiment in Fig 2 onwards, my interpretation was that it was therefore not obvious how to parameterise the model. As such, I was expecting to see some form of uncertainty envelope around the SUGAR model line in Fig 5? How was SUGAR parametrised in this set of runs? I found this very unclear in my head at this point of the manuscript.

In the Caxiuanã simulations, SUGAR was parametrised using the first year of output data from JULES (i.e the year before the panels were put in), together with empirical data as described in the parameter table above. Parameters, $f_{NSC}$, $q_{10}$ and $Y_g$ were parameterised using empirical data or commonly used values. Parameters $\alpha$ and $\phi$ were parametrised using the first year of JULES output. For example, $\alpha$ was found by taking the average CUE over this year. For $a_{K_m}$ we have since conducted a sensitivity study, since this is the least constrained parameter and our updated plots now show an uncertainty envelope based on allowing $a_{K_m}$ to vary between 0.1 and 2.0.

With Fig 5, arguably you don't need panel b, you could perhaps then include a respiration comparison between JULES and SUGAR? No idea what that looks like...

We have include both a comparison of respiration (Page 32) and PCE (Page 31) between SUGAR and JULES as well as the NSC time-series (Page 28) from SUGAR for each plot in the revised manuscript.

$f_{NSC}$: It is a good suggestion to extend the range of tested $f_{NSC}$ values. This was not done originally as there didn't seem to be a clear or scientific way to pick a higher value. Additionally the buffering effect of the model seems to saturate at larger values of $f_{NSC}$ and we felt that the transition from no NSC to 'some' NSC was captured fairly well with the given range. Nonetheless, we have extended the range in the simulations and updated the figures (Figures 3 & 4). We have updated the results section 5.1 accordingly (Page 8 line 40 - page 9 line 4).

$q_{10}$ = 2.0: This is a relatively standard value for Q10 temperature relationships (Ryan 1991) and is commonly used in DGVMs. Changing the temperature dependence in SUGAR would be a relatively easy procedure and so this could be explored in future work, however, for the purposes of this work we felt that using a standard value of 2.0 would be acceptable. We have included this discussion in the updated parameter estimation section (Page 6 line 1).

$a_{K_m}$ is admittedly a not well evaluated parameter, in the sense that it is not evaluated using empirical data. The default value of 0.5 was chosen since it gives numerically stable results (smaller values of $a_{K_m}$ can cause numerical instability with the commonly used minimum time-step of 1 hour) and gives sensible NSC mass fractions. We accept that this is not a scientifically rigorous justification and so have amended the Caxiuanã simulations and conducted a sensitivity study for $a_{K_m}$. The range of $a_{K_m}$ values tested is 0.1-2.0. These figures (Figs. 5 & 6) have replaced the original figures from Caxiuan a (Figs. 5 & 6 of original manuscript).

Pg. 4, lines 11-12. So this means that SUGAR assumes that trees actively allocate to storage to maintain a certain store size. Perhaps acknowledge this decision explicitly here, given the active debate on this (which was introduced in the introduction)?

This section may not be worded particularly well. Currently SUGAR assumes no active storage and NSC content is regulated passively by the asynchrony between GPP and U. The assumption that we are making here is that during non-stressed conditions plants do not rely significantly on their NSC reserves so that over the course of a period of one year, the mass fraction of NSC $\left( \dfrac{C_{NSC}}{C_v} \right)$ remains constant. We then define the parameter $f_{NSC}$ to equal this constant value, which we evaluate using observed NSC mass fractions (Wurth et al 2005). This is then used to determine NSC turnover rate ($\phi$). If over a long period of time these perturbations do not average to zero, as in the TFE simulations at Caxiuanã then the NSC pool will either accumulate or deplete. In this study we look primarily at the first-order effect that this has on ecosystem carbon fluxes (respiration and growth) and have neglected the effects on vegetation dynamics. This is clearly an important process that is the focus of ongoing and future work, but we felt was beyond the scope of this paper. We have updated the model description to make this clearer (Page 4 lines 3-5). Pg. 7, line 8. Is this above-ground NPP or total NPP? If the former, how is the output from SUGAR being adjusted to compensate for this in the evaluation?

Thank you for this comment, it is a good observation as we made a mistake in the original manuscript here. The observations show total NPP except for the root increment component, as this data was not available from the site. The model outputs from SUGAR and JULES are both total NPP and so we have made a correction in Figure 5. For the JULES simulation we

have subtracted simulated root increment ($drootC$) from simulated NPP. Since SUGAR does not distinguish between root, stem, leaf etc we take the fraction $\frac{drootC}{NPP_{total}}$ from JULES and multiply by total simulated NPP in SUGAR. This is then taken from the simulated NPP in SUGAR. Simulated root C increment in JULES is negligible relative to total simulated NPP and so this correction does not qualitatively change our results (see updated figure), but we have updated our results section with the corrected numbers (Page 9 line 56). We have also updated the methods section 4.2.3 (Page 8 lines 29-33) to describe this correction.

Pg. 7, line 16. How is the CV calculated? Annual mean of daily (or monthly) values, followed by taking the mean across the simulated years? Or directly over the whole dataset? I'm trying to understand if this is showing intra- or interannual variability.

CV is calculated as standard deviation/mean so for each gridbox, the standard deviation and mean of simulated PCE are calculated over the entire simulation period which is outputted on a monthly time-step. The mean CV quoted is then the spatial mean across the basin. It is a metric of how variable PCE is over the entire period. Both inter and intra annual variability should be captured by this. To reduce the number of figures in the main text, Figure 2 of the original manuscript has been moved to the appendix and is now Figure A1 (Page 29).

Pg. 8, line 31. Doesn't this imply that the NSC store has been underestimated? Perhaps worth exploring how much storage would be required to maintain respiration and growth and how this is affected by parameters like aKm? This could provide a useful hypothesis for future investigation.

This is essentially what we have tried to explore in the discussion paragraph starting Page 12 line 7 (Pg. 10, line 10. of the original manuscript). Rather than explicitly using SUGAR though, we make an inversion using the observed PCE and predicted GPP by JULES to determine the deficit of carbon required to support growth and respiration.

Pg. 9, lines 42-44. Maybe a bit of over interpretation of small differences here? Overall JULES and SUGAR seem equally good for the control plots in Figs. 5 and 6.

It is true that the models do perform very similarly on the control plot, which is really the main positive result of the control simulations, since JULES is already able to capture control fluxes quite accurately. However, the models do perform differently on shorter time-scales, which we think is still an important result that follows on from the basin level simulation results in section 4.1. Admittedly the original figures (5&6) don't really show this very well. The differences are better illustrated by the new plot of monthly PCE in SUGAR and JULES (Figure A3, Page 31). This figure shows how PCE is buffered against the natural seasonal variation in GPP that the control forest experiences in JULES. The annual averages of PCE within JULES and SUGAR are very similar on the control plot, but as can be seen in this figure, the seasonal variation is different.

Pg. 9, line 51. Whilst the down-regulation of respiration in response to depleting NSC may help buffer NPP, it clearly doesn't improve the simulation of respiration, which is more strongly underestimated in SUGAR than in JULES from 2005 onwards. I think this discussion needs to reflect that whilst SUGAR provides a very useful representation of a process we are confident is important, including this process does not by itself radically improve the overall carbon flux simulation for the drought experiment investigated.

This is a fair comment and we have now include this in our updated discussion section (Page 11 line 43). While SUGAR doesn't improve respiration predictions, it does improve predictions of total PCE for the first 5-6 years of the experiment. The reason that the prediction of respiration is not improved relative to JULES is related to the allocation to growth and respiration in SUGAR. We have shifted the discussion of our results from NPP and respiration to the prediction of total PCE and also included more of a discussion on allocation of C between growth and respiration during drought.

Fig. 5. It would be helpful to also see time-series of respiration and NSC storage to allow a full and balanced interpretation.

We note that this was also requested by the other reviewer and so we have included time-series of predicted NSC in the control and TFE plot at Caxiuanã (Figure. 7, Page 28). We have also included new figures comparing both predicted PCE and respiration from JULES and SUGAR (Figures A3 & A4). The reason for their initial omission was that besides the annual respiration presented in Fig 6., there are no data for R and NSC to evaluate these plots. We agree that to allow a full understanding of how the model is working, it is necessary to have these time-series. However, since we are unable to evaluate these model outputs against observations, and in order to reduce the number of figures in the main text, we have only included the time series of NSC as a main figure and put the PCE and respiration time-series in the appendix.

[revised manuscript text omitted]

---

## Author Response (AR2)

We would like to thank the reviewer for their kind comments at this second stage of review and for spotting the mistakes in the previous manuscript. We have corrected the revisions as outlined below. A marked up manuscript shows these two changes relative to the previous revision. For all the changes relative to the original submission please see the previous author response.

pg 3, line 2. I don't think the sentence has been completed?

I believe this was an editing error and this sentence was supposed to read as it did in the original submission. I have corrected this accordingly.

Pg. 7, line 10. It looks like you've removed the definition of tau? However, it seems good to explicitly define that you mean tau = Cv/Pi_G, as it could also be defined as Cv/NPP for example.

This is a good point, we have put the definition back.

Finally, we would like to thank both reviewers for their constructive feedback throughout the review process which we hope has resulted in a much improved article.

[revised manuscript text omitted]